# MLLM4TS: Leveraging Vision and Multimodal Language Models for General Time-Series Analysis

## Abstract

Effective analysis of time series data presents significant challenges due to the complex temporal dependencies and cross-channel interactions in multivariate data. Inspired by the way human analysts visually inspect time series to uncover hidden patterns, we ask: *can incorporating visual representations enhance automated time-series analysis?* Recent advances in multimodal large language models have demonstrated impressive generalization and visual understanding capability, yet their application to time series remains constrained by the modality gap between continuous numerical data and discrete natural language. To bridge this gap, we introduce MLLM4TS, a novel framework that leverages multimodal large language models for general time-series analysis by integrating a dedicated vision branch. Each time-series channel is rendered as a horizontally stacked color-coded line plot in one composite image to capture spatial dependencies across channels, and a temporal-aware visual patch alignment strategy then aligns visual patches with their corresponding time segments. MLLM4TS fuses fine-grained temporal details from the numerical data with global contextual information derived from the visual representation, providing a unified foundation for multimodal time-series analysis. Extensive experiments on standard benchmarks demonstrate the effectiveness of MLLM4TS across both predictive tasks (e.g., classification) and generative tasks (e.g., anomaly detection and forecasting). These results underscore the potential of integrating visual modalities with pretrained language models to achieve robust and generalizable time-series analysis.

## 1 Introduction

Time-series analysis is a critical task across diverse fields, including manufacturing, finance, healthcare, and environmental monitoring (Hamilton, 2020). It involves monitoring processes (Xu et al., 2022), predicting outcomes (Lim & Zohren, 2021), detecting anomalies (Liu et al., 2024c), and supporting data-driven decision-making (Mahalakshmi et al., 2016). Despite its broad utility, effective analysis remains challenging due to the complex dependencies of sequential data, the integration of multichannel and multimodal signals, and the diversity of tasks requiring application-specific methods. These challenges underscore the need for a unified, generalizable framework that can address a wide range of time-series analysis tasks efficiently and effectively.

Motivated by the observation that analysts often visualize time series to aid interpretation, we consider the role of visual perception in supporting analytical tasks. In anomaly detection as depicted in Figure 1, for instance, anomalies often manifest as visually salient regions - features that visually stand out to enable our eye-brain connection to quickly focus on the most important regions. Similarly, in classification tasks, characteristic motifs or recurring patterns are often preserved in the shape of the time series, serving as indicators of the corresponding class label, much like when a physician examines a patient's electrocardiogram (EKG) for diagnostic purposes. These insights lead to a natural question: *can we mimic human-like visual perception by integrating visual representations into time-series analysis to enhance model performance?*

The emergence of foundation models has initiated a paradigm shift, offering a new lens through which to approach diverse downstream tasks (Bommasani et al., 2022). These models demonstrate remarkably few-shot generalization capabilities, often outperforming task-specific architectures. In

particular, large language models (LLMs) have shown potential for processing and reasoning over data with temporal dependencies (Gruver et al., 2023; Jin et al., 2024), opening up new avenues for advancing time-series analysis. Moreover, recent progress in large vision models has significantly enhanced visual understanding (Dosovitskiy et al., 2021; Liu et al., 2024b), motivating us to explore whether similar capabilities can be leveraged for time-series tasks through visual representation.

However, existing methods for adapting LLMs to time series data often have limitations. One fundamental challenge stems from a modality mismatch: while LLMs are pretrained on discrete token sequences, time series data are inherently continuous, leading to a notable discrepancy (Gruver et al., 2023; Ni et al., 2025). Moreover, many approaches adopt patching strategies that segment time series into smaller chunks (Nie et al., 2022; Wang et al., 2024). Yet, determining an appropriate patch size is non-trivial. If the patches are too large, critical temporal

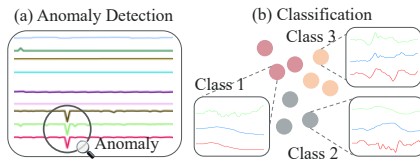

Figure 1: Illustration of the effect of time series visual inspection.

information within each segment may be lost. Conversely, if the patches are too small, the model might overemphasize local features and miss the global temporal patterns in the data. Furthermore, many current methods adopt a channel-independent (Nie et al., 2022; Zhou et al., 2023) approach when dealing with multivariate time series data, neglecting the cross-channel dependencies that recent studies have shown to be essential for capturing the complete dynamics of multivariate time series (Wu et al., 2020; Qiu et al., 2025).

In this paper, we introduce the Multimodal Large Language Model for Time Series (MLLM4TS), a framework that leverages a multimodal foundation model for time-series analysis by utilizing both time series and vision modalities. To address the limitations of language-only models, MLLM4TS introduces a vision branch that transforms multivariate time series into color-coded line-plot images, enabling the capture of global and cross-channel patterns. Visual embeddings derived from a pretrained encoder are then fused with time-series embeddings to jointly model fine-grained temporal dynamics and high-level contextual information. Our contributions are summarized as follows:

- **Modality bridging.** By introducing a vision encoder pretrained for alignment with language-based embeddings (Radford et al., 2021), MLLM4TS effectively bridges the modality gap between continuous time series and discrete natural language, mitigates sensitivity to patch size selection, and enhances its ability to address complex time-series tasks.
- **Temporal-visual alignment.** We introduce a temporal-aware visual patch alignment strategy, which strengthens the alignment between imaged and numerical time series by leveraging the inherent structural properties of time series plots.
- **Versatility and generalization.** The proposed framework demonstrates promising performance across mainstream time series tasks, including classification, anomaly detection, and forecasting, and exhibits robust generalization under few-shot and zero-shot learning settings.

The remainder of this paper is organized as follows: Section 2 provides an overview of related work. Section 3 presents the proposed MLLM4TS framework, detailing its architecture and multimodal fusion approach. Section 4 presents experimental results along with extensive ablation studies to provide deeper insights into the framework's effectiveness. Finally, Section 5 concludes the paper with a summary of findings and directions for future work.

## 2 RELATED WORK

This section reviews prior work in time-series analysis, beginning with traditional methods and followed by recent advances in LLMs, pretrained models, and multimodal learning.

**Traditional Time Series Methods.** Traditional time-series methods, such as ARIMA (Box & Jenkins, 1970), Exponential Smoothing (Hyndman & Athanasopoulos, 2018), and Matrix Profile (Yeh et al., 2016), often struggle with multivariate or high-dimensional data due to their inherent linear assumptions. Deep learning techniques, including RNNs (LSTMs (Hochreiter & Schmidhuber, 1997), GRUs (Cho et al., 2014)), and especially Transformers (Vaswani et al., 2017), have significantly improved performance in complex time series tasks by effectively capturing long-range temporal dependencies, particularly in multivariate contexts (Lim et al., 2021; Wen et al., 2022).

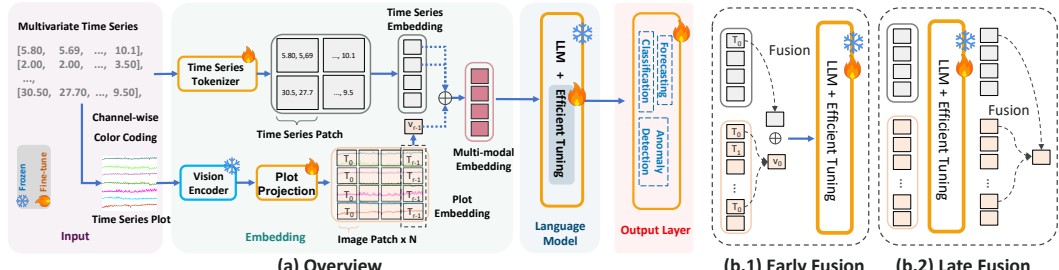

Figure 2: Overview of the MLLM4TS framework. (a) Multivariate time series are tokenized into patches and rendered as colour-coded line plots; the resulting embeddings are fused and passed to a pretrained LLM, followed by a task-specific output head. (b.1) Early fusion combines modalities before LLM processing. (b.2) Late fusion merges them after separate LLM encoding.

**LLMs for Time Series.** The success of LLMs in other domains (Hadi et al., 2023; Liang et al., 2024) has spurred interest in their application to time series. Directly adapting LLMs for time series remains challenging due to the inherent modality gap between continuous time series and discrete language data(Liu et al., 2024e; Tan et al., 2024). Techniques like prompt engineering and patch-based tokenization (Gruver et al., 2023; Nie et al., 2022; Zhou et al., 2023) attempt to bridge this gap, but challenges persist in capturing both global trends and local details, particularly in multivariate time series with complex cross-channel dependencies (Wu et al., 2020; Qiu et al., 2025).

**Pretrained Time Series Models.** Pretrained time-series models have shown promise, particularly in forecasting. Approaches such as Chronos (Ansari et al., 2024) and Moirai (Woo et al., 2024) leverage large-scale data to learn general representations but remain largely restricted to forecasting, with limited effectiveness on tasks like classification and anomaly detection. Similarly, MOMENT (Goswami et al., 2024) adopts a channel-independent design that hinders modeling of cross-channel dependencies. This task- and modality-specific orientation limits their applicability to broader time-series analysis, underscoring the need for more versatile models capable of addressing diverse tasks.

**Multimodal Time-Series Analysis.** Integrating multiple modalities, particularly visual representations, has shown promise in enhancing time-series analysis (Ni et al., 2025). Transforming time series into images, as in ViTST (Li et al., 2024), has proven effective for irregularly sampled time series classification tasks by leveraging pretrained vision transformers (Dosovitskiy et al., 2021). Furthermore, VisionTS (Chen et al., 2025) and Time-VLM (Zhong et al., 2025) utilize vision-language models for few-shot and zero-shot time series forecasting, while TAMA (Zhuang et al., 2024) employs GPT-4o for anomaly detection and interpretation. Nevertheless, these approaches remain specialized for individual downstream tasks, underscoring the need for more general multimodal architectures capable of robustly addressing diverse time-series applications.

Despite these advancements, a unified framework leveraging visual representations across a broad range of time series analysis tasks remains largely unexplored. This work addresses this gap by proposing MLLM4TS, a novel framework that integrates visual embeddings with LLMs to achieve robust and generalizable time series analysis across classification, anomaly detection, and forecasting.

## 3 MLLM4TS FRAMEWORK

The proposed MLLM4TS framework leverages pretrained vision and language models to capture complex temporal dependencies and enable multimodal fusion for time-series analysis. This section outlines the architecture and processing flow of MLLM4TS, as illustrated in Figure 2.

### 3.1 MODEL ARCHITECTURE AND COMPONENTS

The overall MLLM4TS framework comprises four key components: the input module, embedding module, language model, and output layer. We describe each component below.

**(I) Input Module** Given a multivariate time series $\mathbf{x}_{1:L} = \{\mathbf{x}_1, \ldots, \mathbf{x}_L\} \in \mathbb{R}^{L \times C}$ with $L$ time steps and $C$ channels, each channel is converted into a uniquely colored line plot to highlight cross-channel dependencies. These plots are horizontally stacked into a composite image (Figure 10) for the vision encoder, while the raw series is simultaneously input to the time-series tokenizer.

When the number of channels is large, plotting all of them leads to overlap and visual clutter. To address this, we adaptively adjust image size and apply dimensionality reduction by discarding highly correlated channels while preserving representative ones for visualization; the full time series is still maintained in the raw branch. Experiments in the following section show that this enhances plot clarity and that global structure derived from representative subsets remains beneficial.

**(II) Embedding Module**    Each modality is encoded independently to produce embeddings in a shared feature space.

**Time Series Tokenizer.** The time series input is first normalized using reverse instance normalization (Kim et al., 2022), which normalizes the data based on its mean and variance, then added back to the processed output to enhance knowledge transfer. The normalized time series is then partitioned into non-overlapping patches (Nie et al., 2022), allowing the model to capture long-range temporal dependencies with fewer tokens. Finally, a linear projection layer maps each patch to the embedding dimension of the language model for subsequent processing.

**Vision Encoder with Plot Projection.** To model cross-channel dependencies and global patterns, each channel is transformed into a line plot, and the plots are aggregated into a composite image. A pretrained Vision-Language Model (VLM) (Zhang et al., 2024) processes this image to generate embeddings, with the visual encoder kept frozen for stability (Liu et al., 2024b;a). However, since most visual encoders are not pretrained to handle time series data, directly applying them without any adaptation may not achieve optimal performance in time-series applications. To address this, we introduce a plot projection module (a linear transformation) to adapt the visual embeddings for compatibility with the language model. This bridges the gap between channel-specific details and global information by leveraging visual data representations.

**Multimodal Embedding Fusion.** The embeddings generated by the time series tokenizer (time series embedding) and the plot projection (plot embedding) are combined, creating a unified representation of the input data. To better exploit the structural information embedded in time-series plots, we introduce a *Temporal-Aware Visual Patch Alignment* strategy, detailed in Section 3.2. This fusion enables the model to capture complementary information from both modalities, fine-grained temporal patterns and global cross-channel dependencies, thereby enhancing its understanding of complex temporal dynamics. In addition to the fusion strategy itself, we consider two fusion stages. In early fusion (Figure 2(b.1)), the time-series and visual embeddings are combined into multi-modal tokens prior to language model processing. In contrast, late fusion (Figure 2(b.2)) feeds both embeddings into the language model and combines them afterward.

**(III) Language Model**    The core of MLLM4TS consists of a pretrained language model, adapted as a pivot to process embedded multimodal data and understand sequential data through a selective fine-tuning approach. Specifically, the self-attention blocks and Feedforward Neural Network (FNN) layers are kept frozen to retain the generalized knowledge acquired during pretraining. Meanwhile, the positional embeddings and layer normalization layers are fine-tuned, allowing the model to adapt more effectively to the characteristics of time series data. This efficient tuning strategy enables adaptation to new time series tasks with minimal task-specific data, leveraging both preserved pretraining knowledge and targeted fine-tuning (Lu et al., 2022).

**(IV) Output Layer**    The output embeddings from the LLM are passed through a task-specific head to support a range of time series tasks, including classification, anomaly detection, and forecasting. For *classification*, a linear projection followed by a soft-max layer maps the embeddings to a probability distribution over the class set; the model is trained end-to-end with a cross-entropy loss. For *anomaly detection*, the head reconstructs the input sequence $\hat{\mathbf{x}}_{1:L} = \{\hat{\mathbf{x}}_1, \ldots, \hat{\mathbf{x}}_L\} \in \mathbb{R}^{L \times C}$, and an anomaly score is computed from the discrepancy between the original series $\mathbf{x}_{1:L}$ and its reconstruction. For *forecasting*, the head predicts the next $F$ time steps $\mathbf{x}_{L+1:L+F} = \{\mathbf{x}_{L+1}, \ldots, \mathbf{x}_{L+F}\} \in \mathbb{R}^{F \times C}$, where $F$ denotes the forecast horizon. This modular design allows MLLM4TS to flexibly adapt the same backbone to diverse downstream tasks.

## 3.2 Temporal-aware Visual Patch Alignment

To process a two-dimensional line-plot image of a multivariate time series, let the image be $\mathbf{I} \in \mathbb{R}^{H \times W \times C_I}$, where $(H, W)$ is the resolution of the image, $C_I$ is the number of image color channels. The image is partitioned into non-overlapping square patches of size $P \times P$ and processed by a

Vision Transformer (ViT) encoder (Radford et al., 2021). After flattening each patch, we obtain the sequence $\mathbf{Z}_p \in \mathbb{R}^{N \times (P^2 C_I)}$ with $N = HW/P^2$ being the number of patches.

As the time series channels are stacked horizontally, the horizontal axis coincides with absolute time. The number of patches per row is $r = W/P$. Patches sharing the same horizontal index $t \in \{0, \ldots, r-1\}$ correspond to the same time step; we group them as $\mathcal{S}_t = \{\mathbf{Z}_p[t + kr] \mid k = 0, \ldots, q-1\}$, where $q = H/P$. Applying an aggregation function (e.g., average pooling) yields $\mathbf{v}_t = \mathrm{Agg}(\mathcal{S}_t) \in \mathbb{R}^d$, producing the sequence $\{\mathbf{v}_t\}_{t=0}^{r-1}$ that is temporally aligned with the original signal as depicted in Figure 2. This alignment further removes the need for manual patch size tuning by setting the time-series patch size to $L/r$, where $L$ is the time-series length. Finally, we adopt one-dimensional interpolation (upsampling or downsampling) to match the temporal resolution (i.e., length) of the time-series embedding, producing a temporally aligned multimodal representation.

## 4 EXPERIMENTAL ANALYSIS AND DISCUSSION

We conduct a comprehensive evaluation of MLLM4TS across mainstream time-series analysis tasks to address the following research questions (RQs). The key findings are presented in this section, while additional details are provided in the Appendix B.

- **RQ1.** Does incorporating visual representations enhance the performance of general time-series analysis tasks (Section 4.1)?
- **RQ2.** What types of visual representations (e.g., image layouts, visual encoders) are most effective when integrated into the MLLM4TS framework (Section 4.2)?
- **RQ3.** Are language models actually useful for multi-modal time-series analysis (Section 4.3)?

### 4.1 PERFORMANCE OVERVIEW

**A Motivating Example.** Figure 3 presents a motivating example comparing input modalities for time-series classification on five randomly sampled UEA datasets (Bagnall et al., 2018). The multi-modal approach consistently outperforms unimodal baselines, underscoring the importance of integrating local temporal and global contextual information for robust analysis.

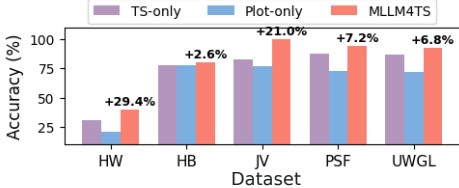

**Main Results.** We evaluate MLLM4TS across three core time-series analysis tasks: classification (Section 4.1.1), anomaly detection (Section 4.1.2), and forecasting (Section 4.1.3), as well as zero-shot learning (Section 4.1.4). For each task, we outline the experimental setup and report results to evaluate the effectiveness of the proposed framework.

Figure 3: Performance comparison of using time series only, plot only, and the multi-modal embeddings.

For fair comparison with the LLM-based TS-only framework OFA (Zhou et al., 2023), we adopt GPT-2 (Radford et al., 2019) as the language model backbone, consistent with prior sequential modeling work. For the vision encoder, we use CLIP-ViT-L-14 (Radford et al., 2021), pretrained for vision-language alignment and well-suited for visualized time-series data. Both MLLM4TS and the reproduced OFA baseline are evaluated over five random runs (error bars in Appendix B.1). Benchmark results are cited from original literature under the same protocol, or reproduced when unavailable. Full experimental details are provided in Appendix A.

#### 4.1.1 TIME-SERIES CLASSIFICATION

**Settings.** For the classification task, we follow the established benchmarking protocols (Zhou et al., 2023; Wu et al., 2023; Gao et al., 2024) on UEA datasets (Bagnall et al., 2018). These datasets, as shown in Table 5, include diverse time series data across domains such as sensor readings, EEG, audio, and speech. Each dataset provides different characteristics in terms of sequence length, number of classes, and data type, offering a comprehensive evaluation in diverse classification scenarios. The model is fine-tuned using cross-entropy loss to minimize classification error, and we apply cross-validation to ensure robust performance estimates.

**Results.** As shown in Figure 4, MLLM4TS outperforms these baselines, highlighting the advantages of its multimodal embedding and fine-tuning strategy.

The combination of time series tokenization and vision-based embeddings helps MLLM4TS capture both local temporal features and global cross-channel dependencies, resulting in improved classification accuracy across most datasets.

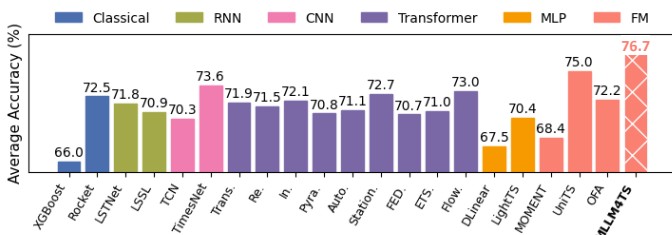

### 4.1.2 TIME-SERIES ANOMALY DETECTION

Figure 4: Model comparison in classification. "*." in the Transformers indicates the name of *former. The results are averaged from 10 subsets of UEA. See Table 8 for full results.

**Settings.** Anomaly detection in time series data is essential for industrial applications, including health monitoring, space exploration, and environmental monitoring. However, progress in evaluating and benchmarking anomaly detection methods has been hindered by issues related to dataset quality, such as mislabeling, bias, and feasibility concerns (Wu & Keogh, 2021; Liu & Paparrizos, 2024). To ensure reliable comparison, we conduct our evaluation using the recently published TSB-AD-M benchmark (Liu & Paparrizos, 2024), a heterogeneous and curated collection comprising 200 multivariate time series (180 for evaluation) from six time series domains. A detailed description of datasets is provided in Table 6.

In addition, to address limitations of traditional evaluation measures - specifically their susceptibility to bias, lack of discrimination and adaptability (Liu & Paparrizos, 2024), we adopt the VUS-PR (Paparrizos et al., 2022; Boniol et al., 2025) as our evaluation measure. VUS-PR improves robustness by reducing sensitivity to temporal lag, enhances accuracy by minimizing bias across different scenarios, and promotes fairness by ensuring consistent evaluations under similar conditions. Additional evaluation results based on the imperfect yet widely employed point-adjusted F score are available in Table 10. For fair comparisons with baseline methods, we use mean squared error (MSE) as the reconstruction loss across all reconstruction-based approaches. During fine-tuning, MLLM4TS is trained to accurately reconstruct normal time series patterns, with anomalies expected to result in higher reconstruction errors.

Table 1: Model comparison in anomaly detection. The top five models from each category in the TSB-AD-M benchmark (Liu & Paparrizos, 2024) are presented, with the detailed evaluation provided in Table 9. VUS-PR is adopted as the primary evaluation measure. Higher values indicate better performance. The best performance is highlighted in bold, and the second-best is underlined.

| Domain | Statistical | | | | | NN | | | | | FM | |
|---|---|---|---|---|---|---|---|---|---|---|---|---|
| | PCA (2017) | KMeansAD (2001) | CBLOF (2003) | MCD (1999) | OCSVM (1999) | CNN (2018) | OmniAnomaly (2019) | LSTMAD (2015) | USAD (2020) | AutoEncoder (2014) | OFA (2023) | MLLM4TS (Ours) |
| Environment | **1.000** | 0.862 | **1.000** | **1.000** | 0.810 | 0.998 | 0.813 | 0.991 | 0.813 | 0.997 | 0.909 | **1.000** |
| Facility | 0.678 | 0.363 | 0.567 | 0.551 | 0.579 | 0.529 | 0.535 | 0.590 | 0.530 | 0.631 | 0.647 | **0.679** |
| Finance | 0.103 | 0.020 | 0.032 | 0.060 | 0.024 | 0.022 | 0.021 | 0.022 | 0.021 | 0.028 | **0.156** | 0.143 |
| HumanActivity | **0.278** | 0.093 | 0.137 | 0.163 | 0.113 | 0.165 | 0.197 | 0.183 | 0.197 | 0.142 | 0.110 | 0.122 |
| Medical | 0.113 | 0.187 | 0.073 | 0.073 | 0.070 | 0.188 | **0.300** | 0.153 | 0.278 | 0.071 | 0.083 | 0.131 |
| Sensor | 0.090 | **0.255** | 0.110 | 0.112 | 0.115 | 0.164 | 0.115 | 0.128 | 0.111 | 0.125 | 0.125 | 0.194 |
| TSB-AD-M | 0.310 | 0.295 | 0.273 | 0.271 | 0.265 | 0.313 | 0.312 | 0.307 | 0.304 | 0.295 | 0.296 | **0.349** |

**Results.** As shown in Table 1, MLLM4TS achieves a substantial improvement over its time-series-only counterpart, OFA, and attains the best overall performance in multivariate time series anomaly detection. It outperforms both statistical and neural network-based baselines, highlighting the effectiveness of introducing vision modality in identifying anomalies in time series.

### 4.1.3 TIME-SERIES FORECASTING

**Settings.** For multivariate time series forecasting, we follow the experimental protocol established by the recent LLM-based forecasting method AutoTimes (Liu et al., 2024e), which incorporates a diverse set of real-world datasets, including ETTh1 (Zhou et al., 2021), ECL, Traffic, Weather (Wu et al., 2021), and Solar-Energy (Liu et al., 2023). Detailed dataset descriptions are provided in the Appendix. To ensure a fair comparison, we adopt GPT-2 as the backbone for AutoTimes and fix the context length $L = 672$ across all baselines. We adopt the "One-for-One" (Liu et al., 2024e)

evaluation across all methods (i.e., training a separate model for each forecasting horizon). Additional discussion on auto-regressive forecasting is provided in Table 16.

Table 2: Model comparison in forecasting. All the results are averaged from 4 different prediction lengths {96, 192, 336, 720}. The best performance is highlighted in bold, and the second-best is underlined. Full results are provided in Table 11.

| Models | MLLM4TS (Ours) | | OFA (2023) | | VisionTS (2025) | | AutoTimes (2024e) | | TimeLLM (2024) | | UniTime (2024d) | | iTrans. (2023) | | DLinear (2023) | | PatchTST (2022) | | TimesNet (2023) | |
|---|---|---|---|---|---|---|---|---|---|---|---|---|---|---|---|---|---|---|---|---|---|
| Metric | MSE | MAE | MSE | MAE | MSE | MAE | MSE | MAE | MSE | MAE | MSE | MAE | MSE | MAE | MSE | MAE | MSE | MAE | MSE | MAE |
| Weather | **0.225** | 0.266 | 0.231 | 0.269 | 0.236 | 0.269 | 0.242 | 0.278 | 0.227 | 0.266 | 0.260 | 0.283 | 0.238 | 0.272 | 0.240 | 0.300 | 0.226 | **0.264** | 0.259 | 0.287 |
| Solar. | **0.188** | 0.246 | 0.229 | 0.296 | 0.231 | 0.266 | 0.197 | **0.242** | 0.234 | 0.293 | 0.254 | 0.291 | 0.202 | 0.269 | 0.217 | 0.278 | 0.189 | 0.257 | 0.200 | 0.268 |
| ETTh1 | 0.408 | 0.430 | 0.426 | 0.438 | 0.398 | **0.415** | **0.397** | 0.425 | 0.409 | 0.432 | 0.438 | 0.445 | 0.438 | 0.450 | 0.423 | 0.437 | 0.413 | 0.431 | 0.458 | 0.450 |
| ECL | 0.165 | 0.261 | 0.167 | 0.264 | **0.157** | **0.251** | 0.173 | 0.266 | 0.170 | 0.275 | 0.194 | 0.287 | 0.161 | 0.256 | 0.177 | 0.274 | 0.159 | 0.253 | 0.192 | 0.295 |
| Traffic | 0.406 | 0.283 | 0.416 | 0.295 | 0.395 | **0.261** | 0.406 | 0.276 | 0.402 | 0.294 | 0.460 | 0.301 | **0.379** | 0.272 | 0.434 | 0.295 | 0.391 | 0.264 | 0.620 | 0.336 |

**Results.** As shown in Table 2, lower MSE and MAE values indicate better forecasting performance. Our multimodal approach achieves competitive results against established baselines, despite their meticulous design and optimization solely for forecasting task. This includes clear advantages on datasets exhibiting periodic patterns, such as Solar-Energy (a showcase provided in Figure 10). Furthermore, reducing the plotted channels to 50 in high-dimensional datasets like Electricity (321 channels) and Traffic (862 channels) consistently improves performance compared to visualizing all channels (details provided in Table 17). This reduction not only enhances clarity and scalability but also shows that global structural information extracted from representative subsets effectively preserves inter-channel interactions for forecasting.

### 4.1.4 FEW/ZERO-SHOT LEARNING

**Settings.** LLMs have demonstrated strong few-shot and zero-shot learning capabilities in natural language tasks (Brown et al., 2020; Kojima et al., 2022). In this section, we investigate whether similar capabilities can be extended to time series analysis. For the few-shot setting, we use only 10% of the available training data to evaluate each model's ability to adapt to data-sparse environments.

For zero-shot learning, we assess the model's capacity for cross-domain generalization: specifically, we evaluate performance on a target dataset $D_A$ without any direct training on it, assuming the model has been trained or pretrained on a different source dataset $D_B$. We compare MLLM4TS with both LLM-based baselines, including OFA (Zhou et al., 2023) and LLMTime (Gruver et al., 2023), as well as recent time series foundation models, such as Chronos (Ansari et al., 2024), Moirai (Woo et al., 2024), and MOMENT (Goswami et al., 2024).

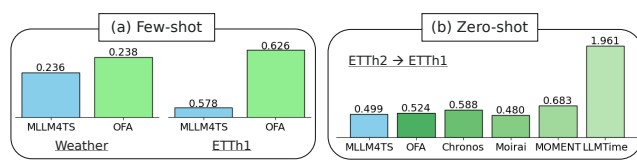

Figure 5: Performance comparison under (a) few-shot and (b) zero-shot settings. Results are reported using MSE (lower the better), averaged across four forecasting horizons {96, 192, 336, 720}. Full results are provided in Table 12 13.

**Results.** As illustrated in Figure 5, MLLM4TS outperforms its time-series-only counterpart under both few-shot and zero-shot learning conditions. In the zero-shot scenario, it also exceeds the performance of time series foundation models that are pretrained exclusively on numerical data, thereby demonstrating superior cross-domain generalization. These findings highlight MLLM4TS's rapid adaptation to previously unseen datasets and resilience to distribution shifts.

### 4.2 VISUAL REPRESENTATION ANALYSIS

With the promising results achieved by our multimodal strategy across mainstream time series analysis tasks, we further investigate the role of different visual representations. This analysis is conducted from four perspectives, as illustrated in Table 3: image layout, visual encoder choice, fusion stage, and sensitivity to patch size selection. For image layout, we compare two configurations: the horizontal layout, where each time series channel is stacked horizontally, and the grid layout, where each channel is plotted within a smaller subregion of the image (a visual example provided in Figure 9). Overall, the horizontal layout consistently outperforms the grid layout, highlighting the effectiveness of our proposed temporal-aware visual patch alignment for aligning visual and

time series modalities. To assess the impact of the visual encoder, we compare two representative architectures: CLIP (Radford et al., 2021), which is pretrained on vision-language alignment tasks, and ResNet (He et al., 2016), which is pretrained solely on image classification. CLIP consistently outperforms ResNet, underscoring its superior capability in processing visual representations of time series due to its alignment with language-based embeddings.

Moreover, we investigate the effectiveness of different fusion strategies, with particular focus on the stage at which modalities are integrated. Experimental results show that early fusion consistently achieves better performance, supporting the hypothesis that **low-level** correlations between imaged time series and numerical time series encode meaningful and complementary information for time-series analysis (Baltrušaitis et al., 2018; Mo

Table 3: Visual representation analysis on image layouts, visual encoders, fusion strategies, and patch-size sensitivity. Accuracy is reported for classification (CLF) and VUS-PR for anomaly detection (AD). Better performance is highlighted in bold. Full results are in Tables 14 and 15.

| Task | Img Layout | | VisualEnc | | FusionStage | | PatchSize STD | |
|---|---|---|---|---|---|---|---|---|
| | Horizontal | Grid | CLIP | ResNet | Early | Late | Plot-TS | TS-Only |
| CLF | **76.7** | 75.2 | **76.7** | 72.6 | **76.7** | 73.5 | **0.56** | 1.13 |
| AD | **0.349** | 0.344 | **0.349** | 0.348 | **0.349** | 0.343 | – | – |

& Morgado, 2024). In addition to its predictive advantages, early fusion exhibits reduced computational cost, as it minimizes the number of tokens that need to be processed by the language model. A detailed runtime comparison is presented in Figure 6.

We further investigate the impact of the vision modality on the model's sensitivity to patch size selection. As shown in the "PatchSize STD" column of Table 3, we report the standard deviation of classification performance across different patch sizes (ranging from $\{1, 2, ..., 10\} \times L/r$, see notation in Section 3.2). Note that for anomaly detection, patch size variation is not applicable, as each input instance corresponds to a fixed-length time series window. Compared to its TS-only counterpart, MLLM4TS exhibits lower performance variance, indicating greater robustness to patch size variation. This stability suggests that the temporal-aware visual alignment preserves the temporal structure more effectively and reduces the model's dependence on precise patch size selection.

## 4.3 LANGUAGE BACKBONE ANALYSIS

With the growing debate over the effectiveness of LLMs for time-series analysis, recent studies have reported that LLM-based methods offer limited advantages over models trained from scratch and fail to adequately capture sequential dependencies in forecasting tasks (Tan et al., 2024). In this work, we extend the scope of investigation to include classification and anomaly detection, examining whether the language modeling capabilities of LLMs are beneficial across a broader range of time series tasks.

As shown in Table 4, we follow the LLM4TS ablation protocol introduced in (Tan et al., 2024), where "LLM" refers to a model using a GPT-2 backbone, and "LLM2Attn" replaces the language model with a single multi-head attention layer (i.e., PAttn, as proposed in the study). In the time-series-only setting, we observe similar trends reported in prior work: replacing the LLM with a simpler attention mechanism results in a 2.6% improvement in forecasting. However, for classification and anomaly detection, models utilizing the full LLM outperform the LLM2Attn variant.

In multimodal scenarios, the benefits of language modeling become more pronounced and consistent, where all three tasks, forecasting, classification, and anomaly detection, benefit from the use of LLMs. This highlights the effectiveness of combining a language-aligned vision encoder with LLMs and underscores the importance of language modeling capabilities for general-purpose multi-modal time series analysis.

Table 4: Comparison of performance between the LLM and LLM2Attn backbones. "Δ Perf" denotes the relative percentage by which LLM outperforms LLM2Attn (positive: LLM better; negative: LLM2Attn better). Results are averaged over 10 UEA datasets for classification (Accuracy), TSB-AD-M for anomaly detection(VUS-PR), and the Weather dataset for forecasting (MSE). See Appendix B.3 for details.

| Task | TS-Only | | | Plot-TS (Ours) | | |
|---|---|---|---|---|---|---|
| | LLM | LLM2Attn | Δ Perf | LLM | LLM2Attn | Δ Perf |
| CLF | 72.2 | 70.2 | 2.80% | 76.7 | 71.4 | 6.90% |
| AD | 0.296 | 0.286 | 3.40% | 0.349 | 0.340 | 2.60% |
| Forecasting | 0.231 | 0.225 | -2.60% | 0.225 | 0.252 | 12.00% |

Despite the promise of language modeling for this task, scaling to billion-parameter models (Touvron et al., 2023; Yang et al., 2024) does not consistently yield improvements over smaller architectures such as GPT-2. This suggests that GPT-2-scale models are sufficiently expressive, while larger models

may introduce issues such as overfitting. Additional results on different language model backbones are provided in Appendix B.3.

We further investigate alternative tuning strategies and their associated runtime implications. As shown in Figure 6(a), the addition of the vision processing branch leads to notable performance gains, albeit at the cost of increased computational overhead. The late fusion strategy incurs a higher runtime due to the longer token sequences passed to the LLM. Furthermore, the TuneAll variants, which involve fine-tuning all model parameters, do not yield improved performance despite their significantly higher computational cost.

We also analyze the training time under different fine-tuning configurations as depicted in Figure 6(b). MLLM4TS adopts the tuning strategy described in Section 3.1, where the "Freeze" variant keeps the pretrained vision and language backbones fixed and updates only the task-specific linear head. In contrast, "TuneVis" further fine-tunes the vision encoder. Among these variants, MLLM4TS achieves the best overall performance while maintaining relatively low training cost, demonstrating the effectiveness of its selective fine-tuning strategy.

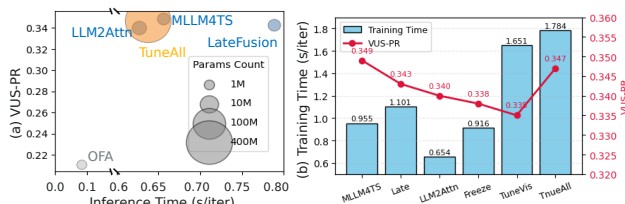

Figure 6: Efficiency evaluation of MLLM4TS and its variants on the TSB-AD benchmark (Liu & Paparrizos, 2024). (a) Inference time versus anomaly-detection performance (VUS-PR), with bubble area proportional to the number of tunable parameters. (b) Training runtime and corresponding VUS-PR, measured using a fixed batch size of 64.

## 4.4 DISCUSSION

We conclude this section by synthesizing research insights in response to the three research questions posed earlier. In this study, we employ a common and intuitive form of time series visualization - the time series line plot. Our findings indicate that incorporating visual representations can significantly enhance the performance of time series analysis by offering an additional modality of data representation, without requiring external information or domain-specific expert knowledge. Through systematic analysis of visual representations, we validate the benefits of exploiting structural patterns embedded in composite line plots and demonstrate the advantage of utilizing visual encoders pretrained on vision-language alignment. The superior performance of early fusion strategies highlights the presence of low-level correlations between imaged and numerical representations of time series data. Furthermore, our results underscore the importance of language modeling capabilities in multimodal time series analysis. The incorporation of a vision modality enhances performance but also introduces additional computational overhead. This observation motivates future work aimed at developing a lightweight visual encoder tailored to time series data, such as pretrained CLIP (Radford et al., 2021) for the time series domain. Overall, this work offers both a conceptual foundation and empirical evidence for the promise of multimodal large language models (Kong et al., 2025; Jiang et al., 2025), particularly in harnessing vision for advanced time series understanding (Ni et al., 2025).

## 5 CONCLUSION

In this paper, we introduced MLLM4TS, a unified multimodal framework for time-series analysis that leverages pretrained language models with vision-based encoders. By combining sequential and visual representations, MLLM4TS captures both local temporal patterns and global cross-channel dependencies, effectively addressing the complexities of multivariate time series. We evaluated its effectiveness on classification, anomaly detection, and forecasting across diverse datasets, demonstrating the complementary contributions of numerical and visual modalities. In summary, MLLM4TS offers a flexible solution for diverse time-series applications, opens avenues for incorporating additional aligned modalities such as images and videos, and motivates future work on lightweight encoders and more effective visual representations.

## REPRODUCIBILITY STATEMENT

We have taken multiple steps to ensure the reproducibility of our work. A full description of the datasets, preprocessing procedures, and benchmark splits is provided in Appendix A.1. The architecture of MLLM4TS is detailed in Section 3, with additional implementation notes and pseudo-code in Appendix A.2. Experimental protocols and evaluation metrics for classification, anomaly detection, forecasting, and zero-/few-shot learning are outlined in Section 4, with further results in Appendix B.

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

# SUPPLEMENTARY MATERIAL FOR MLLM4TS

## A EXPERIMENTAL SETUP

### A.1 DATASET DESCRIPTION

We evaluate MLLM4TS on standard time-series analysis tasks using widely adopted benchmark datasets. For classification and forecasting, we follow the data processing and train-validation-test split protocols established in TimesNet (Wu et al., 2023). For anomaly detection, we use the recent TSB-AD benchmark (Liu & Paparrizos, 2024; Liu et al., 2025), which addresses common concerns regarding the quality and reliability of time-series anomaly detection datasets. Detailed dataset statistics are provided in Table 5 (classification), Table 6 (anomaly detection), and Table 7 (forecasting).

Table 5: Summary of datasets used for the time-series classification task from UEA archive (Bagnall et al., 2018). The table includes the number of training and test samples, sequence length, number of time series dimensions, number of classes, and data type for each dataset. These datasets span various domains such as sensor readings, EEG, audio, and speech, providing a diverse evaluation for time-series classification.

| Dataset | Train | Test | Len | Dim | Class | Domain |
|---|---|---|---|---|---|---|
| EC | 261 | 263 | 1751 | 3 | 4 | SPECTRO |
| FD | 5890 | 3524 | 62 | 144 | 2 | EEG |
| HW | 150 | 850 | 152 | 3 | 26 | HAR |
| HB | 204 | 205 | 405 | 61 | 2 | AUDIO |
| JV | 270 | 370 | 29 | 12 | 9 | AUDIO |
| PSF | 267 | 173 | 144 | 963 | 7 | OTHER |
| SRSCP1 | 268 | 293 | 896 | 6 | 2 | EEG |
| SRSCP2 | 200 | 180 | 1152 | 7 | 2 | EEG |
| SAD | 6599 | 2199 | 93 | 13 | 10 | SPEECH |
| UWGL | 2238 | 2241 | 315 | 3 | 8 | HAR |

Table 6: Summary of datasets used for the time-series anomaly detection task on TSB-AD benchmark (Liu & Paparrizos, 2024). The 'Anomaly Type' column indicates whether the datasets feature point anomalies (P) or sequence anomalies (Seq).

| Domain | Dataset | # TS | Avg Dim | Avg TS Len | Avg # Anomaly | Avg Anomaly Len | Anomaly Ratio | Anomaly Type |
|---|---|---|---|---|---|---|---|---|
| Sensor | GHL (2016) | 25 | 19 | 199001.0 | 2.2 | 1035.2 | 1.1% | Seq |
| | Genesis (2018) | 1 | 18 | 16220.0 | 3.0 | 16.7 | 0.3% | Seq |
| | SWaT (2016) | 2 | 59 | 207457.5 | 16.5 | 1093.6 | 12.7% | Seq |
| | SMAP (2018) | 27 | 25 | 7855.9 | 1.3 | 196.3 | 2.9% | Seq |
| | MSL (2018) | 16 | 55 | 3119.4 | 1.3 | 111.7 | 5.1% | Seq |
| | GECCO (2018) | 1 | 9 | 138521.0 | 51.0 | 33.8 | 1.2% | Seq |
| | CATSv2 (2023) | 6 | 17 | 240000.0 | 11.5 | 811.6 | 3.7% | Seq |
| HumanActivity | Daphnet (2009) | 1 | 9 | 38774.0 | 6.0 | 384.3 | 5.9% | Seq |
| | OPP (2010) | 8 | 248 | 17426.8 | 1.4 | 394.3 | 4.1% | Seq |
| Facility | Exathlon (2021) | 27 | 21 | 60878.4 | 4.3 | 1373.3 | 9.8% | Seq |
| | SMD (2019) | 22 | 38 | 25466.4 | 8.9 | 112.8 | 3.8% | Seq |
| | PSM (2021) | 1 | 25 | 217624.0 | 72.0 | 338.6 | 11.2% | P&Seq |
| Finance | CreditCard (2018) | 1 | 29 | 284807.0 | 465.0 | 1.1 | 0.2% | P&Seq |
| Medical | MITDB (2000) | 13 | 2 | 336153.8 | 15.2 | 1846.8 | 2.7% | Seq |
| | SVDB (1990) | 31 | 2 | 207122.6 | 68.3 | 268.2 | 4.8% | Seq |
| | LTDB (2000) | 5 | 2 | 100000.0 | 105.0 | 134.4 | 15.5% | Seq |
| Environment | TAO (2006) | 13 | 3 | 10000.0 | 788.2 | 1.1 | 8.7% | P&Seq |

Table 7: Summary of datasets used for the time-series forecasting task. 'Dim' denotes the variate number. 'Dataset Size' denotes the total number of time points in (Train, Validation, Test) splits respectively. 'Forecast Length' denotes the future time points to be predicted. 'Frequency' denotes the sampling interval of time points.

| Dataset | Dim | Forecast Length | Dataset Size | Frequency | Domain |
|---|---|---|---|---|---|
| Weather (2021) | 21 | {96, 192, 336, 720} | (36792, 5271, 10540) | 10min | Weather |
| Solar-Energy (2023) | 137 | {96, 192, 336, 720} | (36601, 5161, 10417) | 10min | Energy |
| ETTh1 (2021) | 7 | {96, 192, 336, 720} | (8545, 2881, 2881) | Hourly | Electricity |
| ECL (2021) | 321 | {96, 192, 336, 720} | (18317, 2633, 5261) | Hourly | Electricity |
| Traffic (2021) | 862 | {96, 192, 336, 720} | (12185, 1757, 3509) | Hourly | Transportation |

## A.2 IMPLEMENTATION DETAILS

MLLM4TS converts multivariate time series into a single composite image by plotting each channel as a color-coded line within a horizontally arranged layout. The pseudo-code for this transformation is provided in Algorithm 1, where the resulting image tensor $\hat{I} \in \mathbb{R}^{3 \times H \times W}$ is generated from the input time-series sequence $\mathbf{X} = \{\mathbf{x}_t\}_{t=1}^{L} \in \mathbb{R}^{L \times C}$ and subsequently used in the core model processing.

We present the core processing pipeline of MLLM4TS in Algorithm 2. The image tensor and the original time series are fed into the vision encoder and the time-series tokenization module, respectively, where the latter includes patching and linear projection. The resulting visual and temporal embeddings are aligned using the proposed temporal-aware strategy and subsequently fused before being passed to the language model backbone. The final prediction is obtained via a task-specific linear head on the last-layer hidden states as illustrated in Algorithm 3. For classification, we supervise the predicted logits $\mathbf{Y}_{\text{cls}} \in \mathbb{R}^{B \times K}$ with one-hot labels $\mathbf{Y}^* \in \{0, 1\}^{B \times K}$ via the cross-entropy loss $\mathcal{L}_{\text{cls}} = -\frac{1}{B} \sum_{i=1}^{B} \sum_{k=1}^{K} Y_{i,k}^* \log \left[\text{softmax}(\mathbf{Y}_{\text{cls},i})\right]_k$. For anomaly detection, we reconstruct the input time series $\mathbf{X} \in \mathbb{R}^{B \times L \times C}$ and minimize the mean-squared error $\mathcal{L}_{\text{ad}} = \|\mathbf{Y}_{\text{ad}} - \mathbf{X}\|^2$. The anomaly score is obtained via the reconstruction loss between the original and reconstructed time series. For forecasting, we predict the future series $\mathbf{X}_{L+1:L+F} \in \mathbb{R}^{B \times F \times C}$ and likewise minimize $\mathcal{L}_{\text{fc}} = \|\mathbf{Y}_{\text{fc}} - \mathbf{X}_{L+1:L+F}\|^2$. To handle varying numbers of input channels and enhance generalization, we adopt a cross-channel weight sharing strategy, which implicitly captures inter-variable dependencies during training (Nie et al., 2022). This mechanism complements the visual embeddings that also encode cross-channel relationships.

At the core of MLLM4TS are two pretrained backbones: the vision encoder CLIP-ViT-L-14 (Radford et al., 2021) and the language model GPT-2 (Radford et al., 2019), which are used by default unless stated otherwise. All experiments are conducted using PyTorch on NVIDIA A100 GPUs. We adopt the AdamW optimizer (Loshchilov et al., 2017) with a cosine learning rate scheduler and a warm-up starting at $10^{-6}$. Classification is trained for a maximum of 50 epochs with early stopping patience of 15, while anomaly detection and forecasting use a maximum of 10 epochs with patience set to 3. Trainable projection layers and output heads are implemented as linear layers for simplicity and efficiency. All results are averaged over five runs with different random seeds. Performance stability is illustrated in Figure 7, which includes error bars representing standard deviation.

## B SUPPLEMENTARY RESULTS

In this section, we present supplementary evaluation results for MLLM4TS and baseline methods. Section B.1 provides performance variability illustrated with error bars, followed by comprehensive results for the mainstream time-series analysis tasks in Section B.2. Detailed ablation study results are reported in Section B.3.

---

**Algorithm 1** Time Series to Image

---

**Require:** Input time series $\mathbf{X} = \{\mathbf{x}_t\}_{t=1}^L \in \mathbb{R}^{L \times C}$, image size $(H, W)$
**Ensure:** Image tensor $\hat{I} \in \mathbb{R}^{3 \times H \times W}$
 1: $G \leftarrow (C, 1)$                                                               ▷ grid rows = channels
 2: $colors \leftarrow \mathrm{colormap}(C)$
 3: **for** $c = \{1, \ldots, C\}$ **do**
 4:     $\mathbf{x}^i \leftarrow \mathbf{X}_{:,i}$                                       ▷ extract $i$-th channel
 5:     Create subplot in row $i$ of grid $G$
 6:     $\mathrm{plot}(1 : L, \ \mathbf{x}^i)$ in color $colors[c]$
 7: **end for**
 8: Render the figure to an image tensor $\hat{I}$
 9: **return** $\hat{I}$

---

**Algorithm 2** MLLM4TS Backbone

---

**Require:** Time series $\mathbf{x}_{1:L} \in \mathbb{R}^{L \times C}$, image tensor $\hat{I} \in \mathbb{R}^{3 \times H \times W}$
**Ensure:** Fused multi-modal token features $\mathbf{F} \in \mathbb{R}^{B \times N_{\mathrm{ts}} \times d}$, where $d$ is the feature dimension of language model

 1: **Plot Embedding:**
 2: $\mathbf{V} \leftarrow \mathrm{VisionEncoder}(\hat{I}) \in \mathbb{R}^{B \times N_{\mathrm{vis}} \times d_v}$       ▷ reshape $V \in \mathbb{R}^{d_v \times (B\, N_{\mathrm{vis}})}$ if needed
 3: $\mathbf{V}' \leftarrow W_{\mathrm{proj}} \mathbf{V} + b_{\mathrm{proj}} \in \mathbb{R}^{B \times N_{\mathrm{vis}} \times d}$       ▷ Plot Projection $W_{\mathrm{proj}} \in \mathbb{R}^{d \times d_v}$

 4: **Time Series Embedding:**
 5: $\mu \leftarrow \mathrm{mean}(\mathbf{x}_{1:L}, \ \dim = 1) \in \mathbb{R}^{B \times 1 \times C}$
 6: $\sigma \leftarrow \sqrt{\mathrm{var}(\mathbf{x}_{1:L}, \ \dim = 1) + \epsilon} \in \mathbb{R}^{B \times 1 \times C}$
 7: $\mathbf{x} \leftarrow (\mathbf{x}_{1:L} - \mu) / \sigma$                                    ▷ Time Series Normalization
 8: $\tilde{\mathbf{X}} \leftarrow \mathrm{transpose}(\mathbf{x}_{1:L}, (B, L, C) \rightarrow (B, C, L))$
 9: $\tilde{\mathbf{X}} \leftarrow \mathrm{Padding}(\tilde{\mathbf{X}}) \in \mathbb{R}^{B \times C \times L'}$
10: $\hat{\mathbf{X}} \leftarrow \mathrm{Unfold}(\tilde{\mathbf{X}}, P_{ts}, S) \in \mathbb{R}^{B \times C \times N_{\mathrm{ts}} \times P_{ts}}$       ▷ Patch size $P_{ts} = L/r$ detailed in Section 3.2
11: $\mathbf{T} \leftarrow \mathrm{reshape}(\hat{\mathbf{X}}, (B, N_{\mathrm{ts}}, P_{ts}\, C))$
12: $\mathbf{T}' \leftarrow W_{\mathrm{tok}} \mathbf{T} + b_{\mathrm{tok}} \in \mathbb{R}^{B \times N_{\mathrm{ts}} \times d}$       ▷ Time Series Tokenizer $W_{\mathrm{tok}} \in \mathbb{R}^{d \times P_{ts} C}$

13: **Cross-modal Alignment:**
14: $H = W = \lfloor \sqrt{N_{\mathrm{vis}}} \rfloor$
15: $\mathcal{V} \leftarrow \mathrm{reshape}\big(\mathrm{transpose}(\mathbf{V}', 1, 2), (B, d, H, W)\big)$
16: $\bar{\mathbf{V}} \leftarrow \mathrm{mean}_{\mathrm{H}}(\mathcal{V}) \in \mathbb{R}^{B \times d \times W}$       ▷ Average-pool across the height (H) dimension
17: $\widetilde{\mathbf{V}} \leftarrow \mathrm{interp}(\bar{\mathbf{V}}, N_{\mathrm{ts}}) \in \mathbb{R}^{B \times N_{\mathrm{ts}} \times d}$       ▷ Linear interpolation

18: **Fusion & decoding:**
19: $\mathbf{Z} \leftarrow \widetilde{\mathbf{V}} + \mathbf{T}'$
20: $\mathbf{F} \leftarrow \mathrm{LanguageModel}(\mathbf{Z}) \in \mathbb{R}^{B \times N_{\mathrm{ts}} \times d}$       ▷ Last hidden states
21: **return** $\mathbf{F}$

---

---

**Algorithm 3** Task-Specific Head

---

**Require:** Final hidden states $\mathbf{F} \in \mathbb{R}^{B \times N \times d}$, mean $\boldsymbol{\mu} \in \mathbb{R}^{B \times 1 \times C}$, std. dev. $\boldsymbol{\sigma} \in \mathbb{R}^{B \times 1 \times C}$ from Algorithm 2

**Ensure:** Task outputs $\mathbf{Y}$

1: **Classification head:**
2: $\mathbf{u} \leftarrow \text{Pooling}(\mathbf{F}) \in \mathbb{R}^{B \times d}$
3: $\mathbf{u}' \leftarrow \text{LayerNorm}(\mathbf{u})$
4: $\mathbf{Y}_{\text{cls}} \leftarrow W_{\text{cls}} \mathbf{u}' + b_{\text{cls}} \in \mathbb{R}^{B \times K}$ ▷ Classification logits

5: **Anomaly detection head:**
6: $\mathbf{G} \leftarrow \text{LayerNorm}(\mathbf{F}) \in \mathbb{R}^{B \times L \times d}$ ▷ $L = N$ in anomaly detection
7: $\mathbf{R} \leftarrow W_{\text{ad}} \mathbf{G} + b_{\text{ad}} \in \mathbb{R}^{B \times L \times C}$
8: $\mathbf{Y}_{\text{ad}} \leftarrow \mathbf{R} \times \boldsymbol{\sigma} + \boldsymbol{\mu}$ ▷ Reconstructed time series

9: **Forecasting head:**
10: $\mathbf{H} \leftarrow \text{LayerNorm}(\mathbf{F}) \in \mathbb{R}^{(B \times C) \times (N \times d)}$ ▷ Cross-channel weight sharing mechanism (Nie et al., 2022)
11: $\mathbf{P} \leftarrow W_{\text{fc}} \mathbf{H} + b_{\text{fc}} \in \mathbb{R}^{(B \times C) \times F}$
12: $\mathbf{P}' \leftarrow \text{reshape}(\mathbf{P}, (B, C, F))$
13: $\mathbf{Y}_{\text{fc}} \leftarrow \mathbf{P}' \times \boldsymbol{\sigma} + \boldsymbol{\mu}$ ▷ Predicted time series
14: **return** $\{\mathbf{Y}_{\text{cls}}, \mathbf{Y}_{\text{ad}}, \mathbf{Y}_{\text{fc}}\}$

---

## B.1 ERROR BARS

We report the performance standard deviation of MLLM4TS across five random seeds in Figure 7, based on four evaluation measures available in the TSB-AD benchmark (Liu & Paparrizos, 2024). The consistently low standard deviation across all metrics suggests that MLLM4TS exhibits stable and reliable performance.

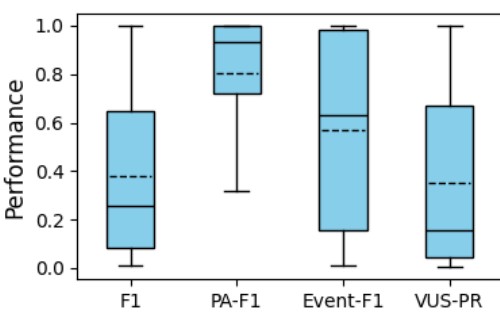 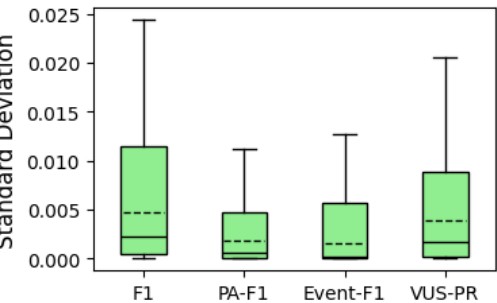

Figure 7: Distribution of standard deviation of four evaluation measures for time-series anomaly detection task on TSB-AD benchmark (comprising 180 time series). Results come from five random seeds. The mean is marked by a dashed line and the median by a solid line.

## B.2 FULL TIME-SERIES ANALYSIS RESULTS

We provide complete evaluation results for time-series classification in Table 8, anomaly detection in Table 9, 10, forecasting in Table 11, few-shot learning in Table 12 and zero-shot learning in Table 13.

## B.3 ABLATION STUDY RESULTS

We present ablation studies across multiple aspects of different time-series analysis tasks, including classification (Table 14), anomaly detection (Table 15), and forecasting (Table 16). In addition, Figure 8 illustrates the effect of different language model backbones, while Table 17 reports the impact of dimensionality reduction during plotting.

Table 8: Performance overview on the classification task on UEA datasets (Bagnall et al., 2018). *. in the Transformers indicates the name of *former. The best performance is highlighted in **bold**, and the second-best is underlined.

| Category | Model | EC | FD | HW | HB | JV | PSF | SRSCP1 | SRSCP2 | SAD | UWGL | Average |
|---|---|---|---|---|---|---|---|---|---|---|---|---|
| Classical | XGBoost (2016) | 43.7 | 63.3 | 15.8 | 73.2 | 86.5 | **98.3** | 84.6 | 48.9 | 69.6 | 75.9 | 66.0 |
| | Rocket (2020) | **45.2** | 64.7 | **58.8** | 75.6 | 96.2 | 75.1 | 90.8 | 53.3 | 71.2 | **94.4** | 72.5 |
| RNN | LSTNet (2018) | 39.9 | 65.7 | 25.8 | 77.1 | 98.1 | 86.7 | 84.0 | 52.8 | **100.0** | 87.8 | 71.8 |
| | LSSL (2021) | 31.1 | 66.7 | 24.6 | 72.7 | 98.4 | 86.1 | 90.8 | 52.2 | 100.0 | 85.9 | 70.9 |
| CNN | TCN (2019) | 28.9 | 52.8 | 53.3 | 75.6 | 98.9 | 68.8 | 84.6 | 55.6 | 95.6 | 88.4 | 70.3 |
| | TimesNet (2023) | 35.7 | 68.6 | 32.1 | 78.0 | 98.4 | 89.6 | 91.8 | 57.2 | 99.0 | 85.3 | 73.6 |
| Transformers | Trans. (2017) | 32.7 | 67.3 | 32.0 | 76.1 | 98.7 | 82.1 | 92.2 | 53.9 | 98.4 | 85.6 | 71.9 |
| | Re. (2020) | 31.9 | 68.6 | 27.4 | 77.1 | 97.8 | 82.7 | 90.4 | 56.7 | 97.0 | 85.6 | 71.5 |
| | In. (2021) | 31.6 | 67.0 | 32.8 | **80.5** | 98.9 | 81.5 | 90.1 | 53.3 | **100.0** | 85.6 | 72.1 |
| | Pyra. (2021) | 30.8 | 65.7 | 29.4 | 75.6 | 98.4 | 83.2 | 88.1 | 53.3 | 99.6 | 83.4 | 70.8 |
| | Auto. (2021) | 31.6 | 68.4 | 36.7 | 74.6 | 96.2 | 82.7 | 84.0 | 50.6 | **100.0** | 85.9 | 71.1 |
| | Station. (2022) | 32.7 | 68.0 | 31.6 | 73.7 | **99.2** | 87.3 | 89.4 | 57.2 | **100.0** | 87.5 | 72.7 |
| | FED. (2022) | 31.2 | 66.0 | 28.0 | 73.7 | 98.4 | 80.9 | 88.7 | 54.4 | **100.0** | 85.3 | 70.7 |
| | ETS. (2022) | 28.1 | 66.3 | 32.5 | 71.2 | 95.9 | 86.0 | 89.6 | 55.0 | **100.0** | 85.0 | 71.0 |
| | Flow. (2022) | 33.8 | 67.6 | 33.8 | 77.6 | 98.9 | 83.8 | 92.5 | 56.1 | 98.8 | 86.6 | 73.0 |
| MLP | DLinear (2023) | 32.6 | 68.0 | 27.0 | 75.1 | 96.2 | 75.1 | 87.3 | 50.5 | 81.4 | 82.1 | 67.5 |
| | LightTS (2022) | 29.7 | 67.5 | 26.1 | 75.1 | 96.2 | 88.4 | 89.8 | 51.1 | **100.0** | 80.3 | 70.4 |
| FM | MOMENT (2024) | 35.7 | 63.3 | 30.8 | 72.2 | 71.6 | 89.6 | 84.0 | 47.8 | 98.1 | 90.9 | 68.4 |
| | UniTS (2024) | 37.6 | **70.5** | 29.7 | 80.0 | 97.8 | 93.1 | **93.9** | **61.1** | 98.9 | 87.8 | 75.0 |
| | OFA (2023) | 33.1 | 69.2 | 30.9 | 78.0 | 82.4 | 87.9 | 93.5 | 60.1 | 99.3 | 86.9 | 72.2 |
| | **MLLM4TS(Ours)** | 38.8 | 68.5 | 40.0 | 80.0 | **99.7** | **94.2** | 93.2 | 60.6 | 99.6 | 92.8 | **76.7** |

Table 9: Performance overview on the anomaly detection task on TSB-AD-M benchmark (Liu & Paparrizos, 2024). Performance is evaluated in VUS-PR (Paparrizos et al., 2022; Boniol et al., 2025). The best performance is highlighted in **bold**, and the second-best is underlined.

| Dataset | Statistical | | | | | NN | | | | | FM | |
|---|---|---|---|---|---|---|---|---|---|---|---|---|
| | PCA (2017) | KMeansAD (2001) | CBLOF (2003) | MCD (1999) | OCSVM (1999) | CNN (2018) | OmniAnomaly (2019) | LSTMAD (2015) | USAD (2020) | AutoEncoder (2014) | OFA (2023) | MLLM4TS (Ours) |
| GHL | 0.012 | 0.030 | 0.019 | 0.014 | 0.036 | 0.062 | **0.065** | 0.062 | **0.065** | 0.047 | 0.007 | 0.007 |
| Genesis | 0.019 | **0.891** | 0.024 | 0.059 | 0.076 | 0.100 | 0.003 | 0.037 | 0.003 | 0.007 | 0.013 | 0.017 |
| SWaT | 0.449 | 0.159 | 0.292 | 0.538 | 0.444 | 0.150 | 0.150 | 0.156 | 0.150 | **0.575** | 0.139 | 0.183 |
| SMAP | 0.093 | **0.380** | 0.137 | 0.104 | 0.116 | 0.193 | 0.124 | 0.163 | 0.108 | 0.129 | 0.208 | 0.348 |
| MSL | 0.149 | **0.435** | 0.215 | 0.229 | 0.216 | 0.217 | 0.217 | 0.217 | 0.226 | 0.219 | 0.212 | 0.237 |
| GECCO | 0.202 | 0.055 | 0.034 | 0.033 | 0.038 | 0.303 | 0.021 | 0.019 | 0.021 | 0.049 | 0.000 | **0.712** |
| CATSv2 | 0.118 | 0.117 | 0.059 | **0.132** | 0.080 | 0.080 | 0.041 | 0.041 | 0.041 | 0.063 | 0.049 | 0.105 |
| Daphnet | 0.130 | 0.297 | 0.096 | 0.135 | 0.064 | 0.203 | 0.340 | 0.311 | 0.340 | 0.129 | **0.378** | 0.338 |
| OPP | **0.299** | 0.063 | 0.143 | 0.167 | 0.121 | 0.177 | 0.177 | 0.165 | 0.177 | 0.144 | 0.072 | 0.091 |
| Exathlon | **0.949** | 0.372 | 0.857 | 0.796 | 0.830 | 0.684 | 0.839 | 0.816 | 0.839 | 0.909 | 0.865 | 0.879 |
| SMD | 0.364 | 0.358 | 0.223 | 0.260 | 0.285 | 0.174 | 0.325 | 0.325 | 0.160 | 0.301 | 0.398 | **0.455** |
| PSM | 0.163 | 0.208 | 0.194 | 0.255 | 0.191 | 0.236 | 0.160 | 0.236 | 0.194 | **0.280** | 0.158 | 0.145 |
| CreditCard | 0.103 | 0.020 | 0.032 | 0.060 | 0.024 | 0.022 | 0.021 | 0.022 | 0.021 | 0.028 | **0.156** | 0.143 |
| MITDB | 0.065 | 0.063 | 0.039 | 0.037 | 0.038 | 0.115 | 0.092 | 0.092 | **0.118** | 0.038 | 0.032 | 0.112 |
| SVDB | 0.112 | 0.203 | 0.068 | 0.067 | 0.065 | **0.352** | 0.155 | 0.155 | 0.322 | 0.065 | 0.096 | 0.117 |
| LTDB | 0.244 | 0.414 | 0.202 | 0.214 | 0.198 | 0.303 | **0.444** | 0.303 | 0.411 | 0.206 | 0.134 | 0.287 |
| TAO | **1.000** | 0.862 | **1.000** | **1.000** | 0.810 | 0.991 | 0.813 | 0.991 | 0.813 | 0.997 | 0.909 | **1.000** |
| **TSB-AD-M** | 0.310 | 0.295 | 0.273 | 0.271 | 0.265 | 0.312 | 0.312 | 0.307 | 0.304 | 0.295 | 0.296 | **0.349** |

Table 10: Performance overview on the anomaly detection task on four common datasets. Performance is evaluated in point-adjusted F-score. The best performance is highlighted in **bold**, and the second-best is underlined.

| Dataset | MLLM4TS (Ours) | OFA (2023) | TimesNet (2023) | PatchTS. (2022) | ETS. (2022) | FED. (2022) | LightTS (2022) | DLinear (2023) | Station. (2022) | Auto. (2021) | Pyra. (2021) | In. (2021) | Re. (2020) | Trans. (2017) |
|---|---|---|---|---|---|---|---|---|---|---|---|---|---|---|
| SMD | **87.4** | 86.9 | 84.6 | 84.6 | 83.1 | 85.1 | 82.5 | 77.1 | 84.7 | 85.1 | 83.0 | 81.7 | 75.3 | 79.6 |
| MSL | **90.8** | 81.8 | 81.8 | 78.7 | 85.0 | 78.6 | 79.0 | 84.9 | 77.5 | 79.1 | 84.9 | 84.1 | 84.4 | 78.7 |
| SMAP | **78.4** | 68.8 | 69.4 | 68.8 | 69.5 | 70.8 | 69.2 | 69.3 | 71.1 | 71.1 | 71.1 | 69.9 | 70.4 | 69.7 |
| SWaT | **95.5** | 95.1 | 93.0 | 85.7 | 84.9 | 93.2 | 93.3 | 87.5 | 79.9 | 92.7 | 91.8 | 81.4 | 82.8 | 80.4 |
| PSM | **97.6** | 97.1 | 97.3 | 96.1 | 91.8 | 97.2 | 97.2 | 93.6 | 97.3 | 93.3 | 82.1 | 77.1 | 73.6 | 76.1 |
| Average | **89.9** | 85.9 | 85.2 | 82.8 | 82.9 | 85.0 | 84.2 | 82.5 | 82.1 | 84.3 | 82.6 | 78.8 | 77.3 | 76.9 |

Table 11: Performance overview on the forecasting task. The best performance is highlighted in **bold**, and the second-best is underlined.

| Method | | MLLM4TS (Ours) | | OFA (2023) | | VisionTS (2025) | | AutoTimes (2024e) | | TimeLLM (2024) | | UniTime (2024d) | | iTrans. (2023) | | DLinear (2023) | | PatchTST (2022) | | TimesNet (2023) | |
|---|---|---|---|---|---|---|---|---|---|---|---|---|---|---|---|---|---|---|---|---|---|
| Metric | | MSE | MAE | MSE | MAE | MSE | MAE | MSE | MAE | MSE | MAE | MSE | MAE | MSE | MAE | MSE | MAE | MSE | MAE | MSE | MAE |
| Weather | 96 | 0.149 | 0.198 | 0.154 | 0.205 | **0.144** | **0.196** | 0.158 | 0.208 | 0.149 | 0.200 | 0.180 | 0.223 | 0.163 | 0.211 | 0.152 | 0.237 | 0.149 | 0.198 | 0.172 | 0.220 |
| | 192 | **0.193** | 0.245 | 0.196 | 0.245 | 0.196 | 0.243 | 0.207 | 0.254 | 0.195 | 0.243 | 0.205 | 0.261 | 0.205 | 0.282 | 0.220 | 0.282 | 0.194 | **0.241** | 0.219 | 0.261 |
| | 336 | **0.243** | **0.282** | 0.254 | 0.290 | 0.265 | 0.295 | 0.262 | 0.298 | 0.245 | **0.282** | 0.280 | 0.300 | 0.254 | 0.289 | 0.265 | 0.319 | 0.245 | **0.282** | 0.280 | 0.306 |
| | 720 | 0.315 | 0.337 | 0.321 | 0.337 | 0.337 | 0.342 | 0.342 | 0.353 | 0.318 | 0.338 | 0.355 | 0.348 | 0.329 | 0.340 | 0.323 | 0.362 | **0.314** | **0.334** | 0.365 | 0.359 |
| | Avg | **0.225** | 0.266 | 0.231 | 0.269 | 0.236 | 0.269 | 0.242 | 0.278 | 0.227 | 0.266 | 0.260 | 0.283 | 0.238 | 0.272 | 0.240 | 0.300 | 0.226 | **0.264** | 0.259 | 0.287 |
| Solar | 96 | **0.167** | 0.231 | 0.196 | 0.261 | 0.213 | 0.241 | 0.171 | **0.221** | 0.224 | 0.289 | 0.223 | 0.274 | 0.187 | 0.255 | 0.191 | 0.256 | 0.168 | 0.237 | 0.178 | 0.256 |
| | 192 | **0.185** | 0.245 | 0.224 | 0.292 | 0.233 | 0.262 | 0.190 | **0.236** | 0.244 | 0.289 | 0.251 | 0.290 | 0.200 | 0.270 | 0.211 | 0.273 | 0.187 | 0.263 | 0.200 | 0.268 |
| | 336 | **0.192** | 0.251 | 0.240 | 0.308 | 0.236 | 0.270 | 0.203 | **0.248** | 0.225 | 0.270 | 0.270 | 0.301 | 0.209 | 0.276 | 0.228 | 0.287 | 0.196 | 0.260 | 0.212 | 0.274 |
| | 720 | 0.209 | **0.257** | 0.256 | 0.321 | 0.241 | 0.289 | 0.222 | 0.262 | 0.243 | 0.301 | 0.271 | 0.298 | 0.213 | 0.276 | 0.236 | 0.295 | **0.205** | 0.269 | 0.211 | 0.273 |
| | Avg | **0.188** | 0.246 | 0.229 | 0.296 | 0.231 | 0.266 | 0.197 | **0.242** | 0.234 | 0.293 | 0.254 | 0.291 | 0.202 | 0.269 | 0.217 | 0.278 | 0.189 | 0.257 | 0.200 | 0.268 |
| ETTh1 | 96 | 0.366 | 0.400 | 0.377 | 0.404 | **0.343** | **0.376** | 0.360 | 0.397 | 0.380 | 0.412 | 0.386 | 0.409 | 0.386 | 0.405 | 0.375 | 0.399 | 0.370 | 0.399 | 0.384 | 0.402 |
| | 192 | 0.404 | 0.420 | 0.413 | 0.424 | **0.379** | **0.405** | 0.391 | 0.419 | 0.405 | 0.422 | 0.428 | 0.436 | 0.422 | 0.439 | 0.405 | 0.416 | 0.413 | 0.421 | 0.557 | 0.436 |
| | 336 | 0.425 | 0.434 | 0.436 | 0.444 | **0.412** | **0.423** | 0.408 | 0.432 | 0.422 | 0.433 | 0.464 | 0.456 | 0.444 | 0.457 | 0.439 | 0.443 | 0.422 | 0.436 | 0.491 | 0.469 |
| | 720 | 0.436 | 0.467 | 0.477 | 0.481 | 0.458 | 0.455 | **0.429** | **0.452** | 0.430 | 0.459 | 0.473 | 0.479 | 0.500 | 0.498 | 0.472 | 0.490 | 0.447 | 0.466 | 0.521 | 0.500 |
| | Avg | 0.408 | 0.430 | 0.426 | 0.438 | 0.398 | 0.415 | **0.397** | 0.425 | 0.409 | 0.432 | 0.438 | 0.445 | 0.438 | 0.450 | 0.423 | 0.437 | 0.413 | 0.431 | 0.458 | 0.450 |
| ECL | 96 | 0.134 | 0.232 | 0.137 | 0.236 | **0.126** | 0.223 | 0.140 | 0.236 | 0.137 | 0.244 | 0.171 | 0.266 | 0.132 | 0.227 | 0.153 | 0.237 | 0.129 | **0.222** | 0.168 | 0.272 |
| | 192 | 0.153 | 0.251 | 0.154 | 0.251 | **0.144** | 0.241 | 0.159 | 0.253 | 0.162 | 0.271 | 0.178 | 0.274 | 0.153 | 0.249 | 0.152 | 0.249 | 0.147 | **0.240** | 0.184 | 0.289 |
| | 336 | 0.169 | 0.267 | 0.169 | 0.267 | **0.163** | **0.255** | 0.177 | 0.270 | 0.175 | 0.279 | 0.194 | 0.289 | 0.167 | 0.262 | 0.169 | 0.267 | 0.163 | 0.259 | 0.198 | 0.300 |
| | 720 | 0.204 | 0.295 | 0.207 | 0.300 | **0.195** | **0.286** | 0.216 | 0.303 | 0.207 | 0.303 | 0.232 | 0.319 | 0.192 | 0.285 | 0.233 | 0.344 | 0.197 | 0.290 | 0.220 | 0.320 |
| | Avg | 0.165 | 0.261 | 0.167 | 0.264 | **0.157** | **0.251** | 0.173 | 0.266 | 0.170 | 0.275 | 0.194 | 0.287 | 0.161 | 0.256 | 0.177 | 0.274 | 0.159 | 0.253 | 0.192 | 0.295 |
| Traffic | 96 | 0.378 | 0.268 | 0.395 | 0.283 | 0.356 | **0.245** | 0.369 | 0.257 | 0.373 | 0.280 | 0.438 | 0.291 | **0.351** | 0.257 | 0.410 | 0.282 | 0.360 | 0.249 | 0.593 | 0.321 |
| | 192 | 0.396 | 0.281 | 0.410 | 0.290 | 0.385 | **0.251** | 0.394 | 0.268 | 0.390 | 0.288 | 0.446 | 0.293 | **0.364** | 0.265 | 0.423 | 0.287 | 0.379 | 0.256 | 0.617 | 0.336 |
| | 336 | 0.404 | 0.280 | 0.414 | 0.295 | 0.398 | **0.262** | 0.413 | 0.278 | 0.407 | 0.299 | 0.461 | 0.300 | **0.382** | 0.273 | 0.436 | 0.296 | 0.392 | 0.264 | 0.629 | 0.336 |
| | 720 | 0.446 | 0.304 | 0.445 | 0.311 | 0.439 | **0.284** | 0.449 | 0.299 | 0.438 | 0.310 | 0.494 | 0.318 | **0.420** | 0.292 | 0.466 | 0.315 | 0.432 | 0.286 | 0.640 | 0.350 |
| | Avg | 0.406 | 0.283 | 0.416 | 0.295 | 0.395 | **0.261** | 0.406 | 0.276 | 0.402 | 0.294 | 0.460 | 0.301 | **0.379** | 0.272 | 0.434 | 0.295 | 0.391 | 0.264 | 0.620 | 0.336 |

Table 12: Performance comparison between MLLM4TS and OFA (Zhou et al., 2023) on few-shot forecasting. 10% of the training data is used to train the model. We mark the better performance in **bold**.

| Method | Horizon | MLLM4TS | | OFA | |
|---|---|---|---|---|---|
| | Metric | MSE | MAE | MSE | MAE |
| Weather | 96 | 0.164 | 0.219 | **0.161** | **0.212** |
| | 192 | **0.202** | **0.247** | 0.207 | 0.253 |
| | 336 | **0.258** | **0.295** | 0.264 | 0.298 |
| | 720 | 0.322 | 0.338 | **0.321** | **0.335** |
| | Avg | **0.236** | **0.274** | 0.238 | 0.275 |
| ETTh1 | 96 | 0.494 | 0.489 | **0.464** | **0.472** |
| | 192 | **0.522** | 0.509 | 0.526 | **0.507** |
| | 336 | **0.534** | **0.511** | 0.747 | 0.601 |
| | 720 | **0.761** | 0.633 | 0.769 | **0.632** |
| | Avg | **0.578** | **0.535** | 0.626 | 0.553 |

Table 13: Performance overview on zero-shot forecasting. For MLLM4TS and OFA, the model is trained on ETTh2 dataset and then tested on ETTh1 dataset. For time series foundation models pretrained from time series corpus, the models are directly applied on ETTh1 dataset. The best performance is highlighted in **bold**, and the second-best is underlined.

| Dataset | Horizon | MLLM4TS | | OFA (2023) | | Chronos (2024) | | Moirai (2024) | | MOMENT (2024) | | LLMTime (2023) | |
|---|---|---|---|---|---|---|---|---|---|---|---|---|---|
| | | MSE | MAE | MSE | MAE | MSE | MAE | MSE | MAE | MSE | MAE | MSE | MAE |
| ETTh2 → ETTh1 | 96 | 0.503 | 0.488 | 0.459 | 0.458 | 0.441 | 0.390 | **0.381** | **0.388** | 0.688 | 0.557 | 1.130 | 0.777 |
| | 192 | 0.454 | 0.459 | 0.496 | 0.481 | 0.502 | 0.524 | **0.434** | **0.415** | 0.688 | 0.560 | 1.242 | 0.820 |
| | 336 | 0.518 | 0.496 | 0.537 | 0.517 | 0.576 | 0.467 | **0.485** | **0.445** | 0.675 | 0.563 | 1.328 | 0.864 |
| | 720 | **0.521** | 0.517 | 0.604 | 0.556 | 0.835 | 0.583 | 0.611 | **0.510** | 0.683 | 0.585 | 4.145 | 1.461 |
| | Avg | 0.499 | 0.490 | 0.524 | 0.503 | 0.588 | 0.466 | **0.480** | **0.430** | 0.683 | 0.560 | 1.961 | 0.981 |

The consistent performance improvements over the time-series-only counterpart underscore the effectiveness of incorporating a vision modality into time-series analysis. Ablation results across different model variants further validate the robustness of the proposed framework. An alternative to LLM-based direct forecasting is the autoregressive approach, where future values are generated sequentially based on previously predicted outputs, as suggested by recent work (Liu et al., 2024e). As shown in Table 16 ('AutoReg'), this method performs well for short prediction horizons but suffers from error accumulation as the forecasting window lengthens, leading to degraded performance. Moreover, increasing the language model size from GPT-2 (Radford et al., 2019) (124M parameters) to larger models such as Qwen3 (Yang et al., 2024) (1.7B parameters) does not yield further gains in multimodal time-series tasks. These findings suggest that smaller models provide sufficient language modeling capacity, while larger models may be more prone to overfitting noise in time-series data, potentially hindering generalization.

Table 14: Ablation study on time-series classification task. 'LLM2Attn' replaces the language model with one single attention layer. 'Layout' replaces horizontal layout with grid layout. 'VisualEnc' replaces CLIP with ResNet. 'Fusion' replaces early fusion with late fusion.

| Dataset | TS-Only | | Plot-TS | | | | |
|---|---|---|---|---|---|---|---|
| | OFA | LLM2Attn | MLLM4TS | Layout | VisualEnc | Fusion | LLM2Attn |
| EC | 33.1 | 33.1 | 38.8 | 41.1 | 34.1 | 35.0 | 33.8 |
| FD | 69.2 | 66.1 | 68.5 | 58.7 | 68.6 | 58.0 | 60.2 |
| HW | 30.9 | 32.9 | 40.0 | 43.1 | 34.2 | 33.9 | 32.6 |
| HB | 78.0 | 75.6 | 80.0 | 80.5 | 78.1 | 80.0 | 77.6 |
| JV | 82.4 | 93.8 | 99.7 | 99.2 | 98.7 | 99.2 | 98.4 |
| PSF | 87.9 | 89.0 | 94.2 | 89.0 | 85.6 | 86.7 | 83.8 |
| SRSCP1 | 93.5 | 93.2 | 93.2 | 91.1 | 90.1 | 89.8 | 91.8 |
| SRSCP2 | 60.1 | 57.2 | 60.6 | 62.2 | 57.2 | 64.4 | 56.7 |
| SAD | 99.3 | 73.8 | 99.6 | 99.1 | 87.7 | 96.3 | 87.3 |
| UWGL | 86.9 | 87.2 | 92.8 | 87.9 | 91.9 | 91.3 | 91.6 |
| Average | 72.2 | 70.2 | 76.7 | 75.2 | 72.6 | 73.5 | 71.4 |

Table 15: Ablation study on time-series anomaly detection task. 'LLM2Attn' replaces the language model with one single attention layer. 'Layout' replaces horizontal layout with grid layout. 'VisualEnc' replaces CLIP with ResNet. 'Fusion' replaces early fusion with late fusion.

| Domain | TS-Only | | Plot-TS | | | | |
|---|---|---|---|---|---|---|---|
| | OFA | LLM2Attn | MLLM4TS | Layout | VisualEnc | Fusion | LLM2Attn |
| Environment | 0.909 | 1.000 | 1.000 | 1.000 | 1.000 | 1.000 | 0.886 |
| Facility | 0.647 | 0.599 | 0.679 | 0.678 | 0.692 | 0.679 | 0.692 |
| Finance | 0.156 | 0.154 | 0.143 | 0.145 | 0.145 | 0.149 | 0.222 |
| HumanActivity | 0.110 | 0.102 | 0.122 | 0.120 | 0.122 | 0.120 | 0.092 |
| Medical | 0.083 | 0.085 | 0.131 | 0.134 | 0.139 | 0.133 | 0.163 |
| Sensor | 0.125 | 0.117 | 0.194 | 0.180 | 0.181 | 0.179 | 0.164 |
| Average | 0.296 | 0.286 | 0.349 | 0.344 | 0.348 | 0.343 | 0.340 |

## C  SHOW CASE

We provide example time series plots in Section C.1 and an illustration of the attention map for multi-modal tokens in language models in Section C.2.

### C.1  EXAMPLE PLOTS

We provide an illustration of horizontal and grid layout in Figure 9, and example plots of datasets used in this work in Figure 10.

Table 16: Ablation study on time-series forecasting task. 'AutoReg' trains the model and generates forecasting results in an autoregressive manner as described in AutoTimes (Liu et al., 2024e). 'VisualEnc' replaces CLIP with ResNet. 'Fusion' replaces early fusion with late fusion. 'LLM2Attn' replaces the language model with one single attention layer.

| Dataset | ETTh1 | | | | | | | | Weather | | | | | | | |
|---|---|---|---|---|---|---|---|---|---|---|---|---|---|---|---|---|
| Type | MLLM4TS | | AutoReg | | VisualEnc | | LLM2Attn | | MLLM4TS | | AutoReg | | VisualEnc | | LLM2Attn | |
| Metric | MSE | MAE | MSE | MAE | MSE | MAE | MSE | MAE | MSE | MAE | MSE | MAE | MSE | MAE | MSE | MAE |
| Pred-96 | 0.366 | 0.4 | **0.361** | **0.394** | 0.397 | 0.423 | 0.479 | 0.483 | **0.149** | **0.198** | **0.149** | 0.201 | 0.165 | 0.217 | 0.187 | 0.243 |
| Pred-192 | 0.404 | 0.42 | **0.397** | **0.417** | 0.436 | 0.593 | 0.495 | 0.509 | **0.193** | **0.245** | 0.202 | 0.248 | 0.227 | 0.271 | 0.225 | 0.273 |
| Pred-336 | 0.425 | 0.434 | **0.42** | **0.433** | 0.475 | 0.477 | 0.506 | 0.522 | **0.243** | **0.282** | 0.263 | 0.291 | 0.269 | 0.307 | 0.267 | 0.303 |
| Pred-720 | **0.436** | 0.467 | 0.446 | **0.46** | 0.515 | 0.514 | 0.540 | 0.561 | **0.315** | **0.337** | 0.336 | 0.343 | 0.331 | 0.352 | 0.328 | 0.347 |

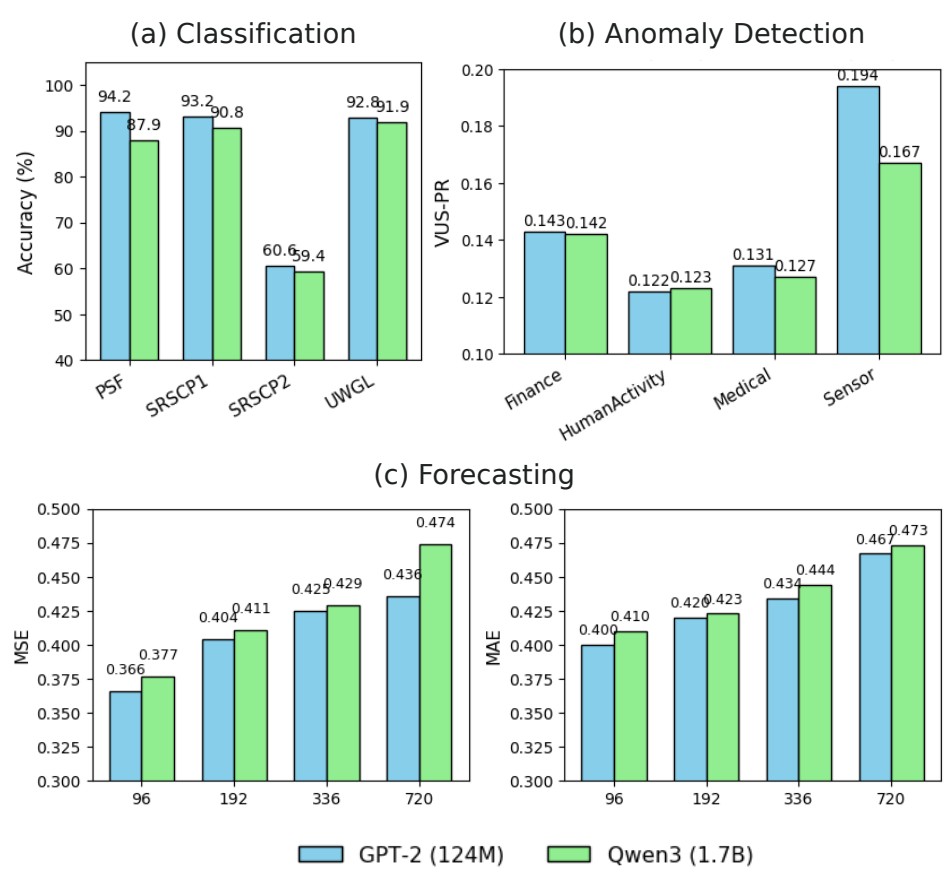

Figure 8: Comparison of task performance using different language model backbones: GPT-2 (Radford et al., 2019) (124M parameters) and Qwen3 (Yang et al., 2024) (1.7B parameters). Panels (a) classification and (b) anomaly detection show metrics for which higher values indicate better performance, while panel (c) forecasting presents forecasting errors, where lower values denote better results.

Table 17: Impact of Dimensionality Reduction. MLLM4TS (Dim.) refers to reducing the plotted channels to 50 by removing highly correlated time-series channels, whereas MLLM4TS (Orig.) denotes visualization using all original channels.

| Method | Horizon | MLLM4TS (Dim.) | | MLLM4TS (Orig.) | |
|---|---|---|---|---|---|
| | Metric | MSE | MAE | MSE | MAE |
| ECL | 96 | 0.134 | 0.232 | 0.141 | 0.244 |
| | 192 | 0.153 | 0.251 | 0.157 | 0.253 |
| | 336 | 0.169 | 0.267 | 0.168 | 0.263 |
| | 720 | 0.204 | 0.295 | 0.201 | 0.293 |
| | Avg | 0.165 | 0.261 | 0.166 | 0.263 |
| Traffic | 96 | 0.378 | 0.268 | 0.386 | 0.281 |
| | 192 | 0.396 | 0.281 | 0.399 | 0.289 |
| | 336 | 0.404 | 0.28 | 0.420 | 0.304 |
| | 720 | 0.446 | 0.304 | 0.449 | 0.308 |
| | Avg | 0.406 | 0.283 | 0.414 | 0.295 |

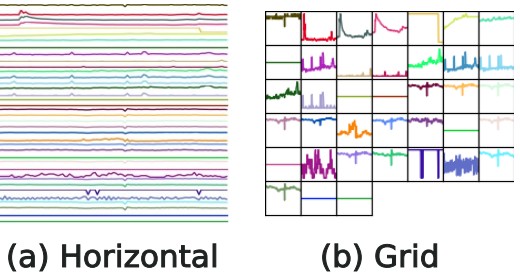

(a) Horizontal            (b) Grid

Figure 9: Example plots of (a) horizontal and (b) grid layout.

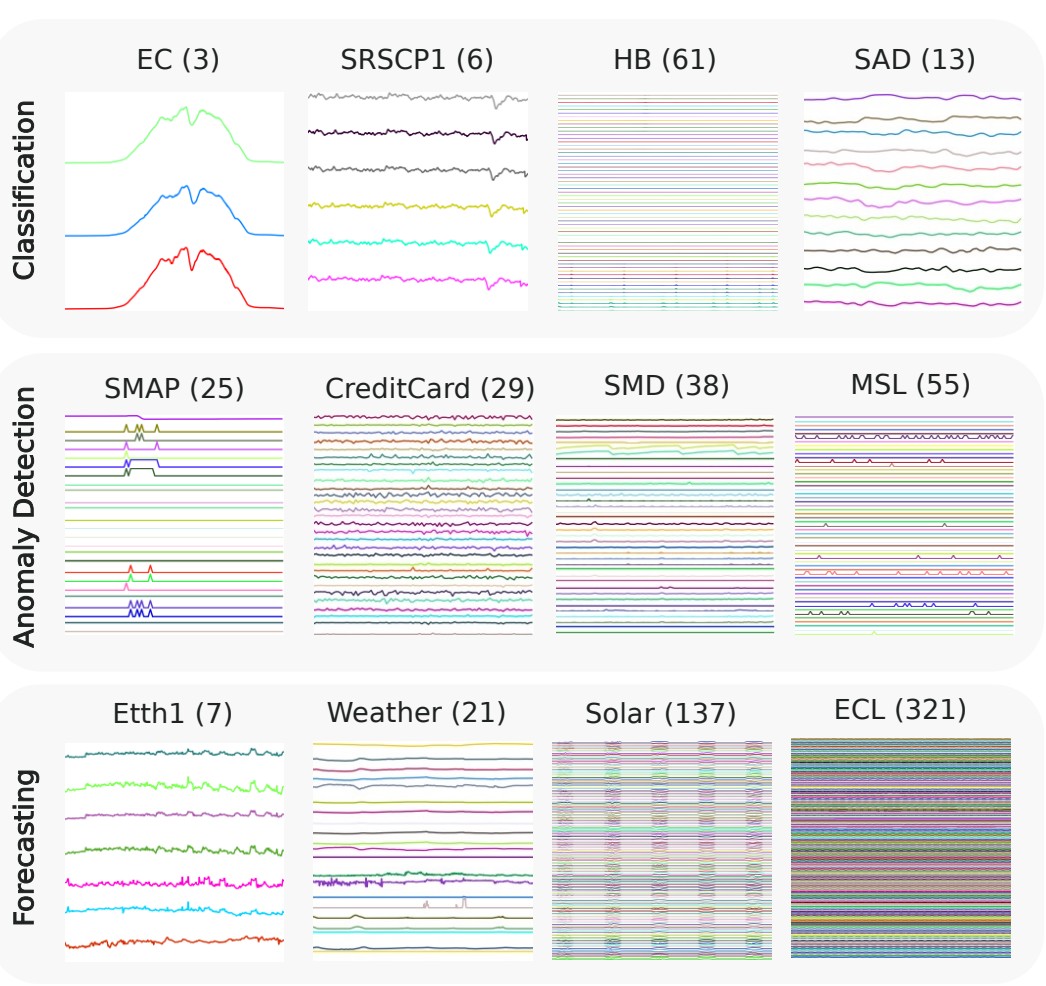

Figure 10: Example plots of an instance from datasets used in this work (dataset name and channel count in parentheses).

## C.2 ATTENTION MAPS

To better understand how the language model processes multimodal input, we visualize the attention maps of the language model backbone. The input consists of a concatenation of time-series tokens $(\text{TS}_1, \ldots, \text{TS}_N)$ and visual tokens $(\text{V}_1, \ldots, \text{V}_M)$. In the early transformer layers, attention is primarily concentrated on the time-series tokens, with limited attention directed toward the visual tokens. As the depth of the model increases, attention becomes more evenly distributed across both modalities, indicating that cross-modal interactions become more prominent in the deeper layers of the language model.

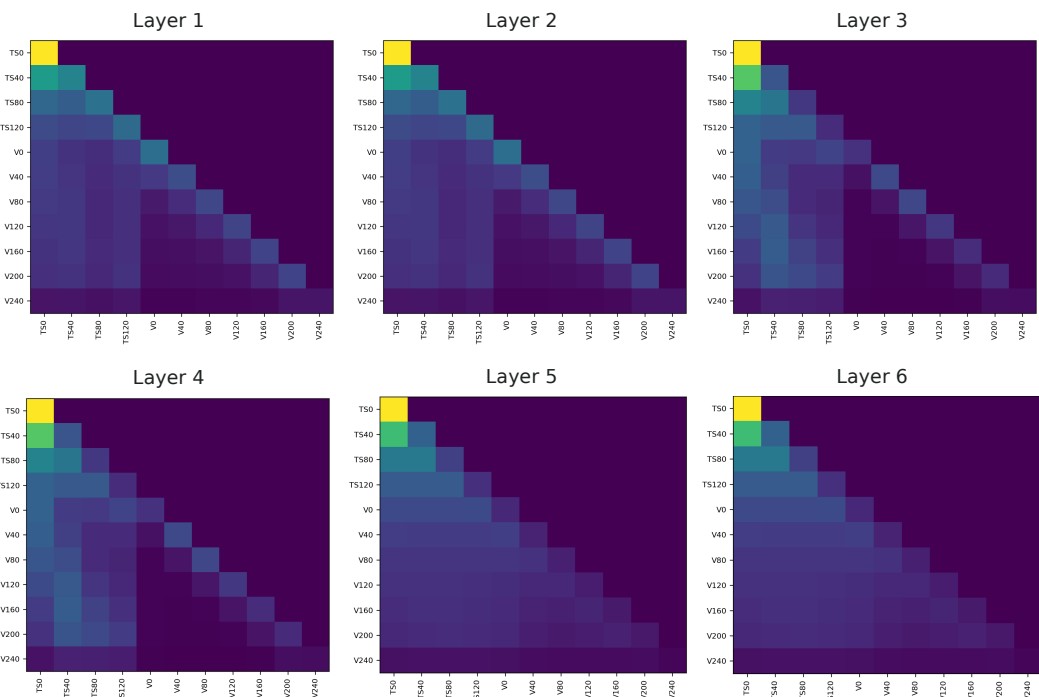

Figure 11: Layer-wise self-attention maps of the language model in MLLM4TS (Late) for the classification task. Each heatmap displays the average attention weights over the entire test set. The input tokens are concatenated time-series tokens $(\text{TS}_1, \ldots, \text{TS}_N)$ and visual tokens $(\text{V}_1, \ldots, \text{V}_M)$.

## C.3 IMPACT OF CHANNEL COLOR CODING

We further investigate the impact of channel-specific color coding in time series plots. As illustrated in Figure 12, we compare three configurations: (1) MLLM4TS with color-coded channels, (2) MLLM4TS without color coding, and (3) a time-series-only baseline. The version of MLLM4TS without color coding achieves intermediate performance between the other two settings. This suggests that while the inclusion of visual representations alone provides benefits, applying channel-wise color coding enhances the model's ability to capture cross-channel dependencies, demonstrating the effectiveness of the proposed design.

## D BROADER IMPACT

**Impact on Real-world Applications.** MLLM4TS offers a unified and effective solution for a wide range of time series analysis tasks, including but not limited to classification, anomaly detection, and forecasting. These capabilities support practical applications in domains such as electrocardiogram monitoring, human activity recognition, financial modeling, facility management, environmental sensing, and industrial process control. Its strong performance in few-shot and zero-shot scenarios,

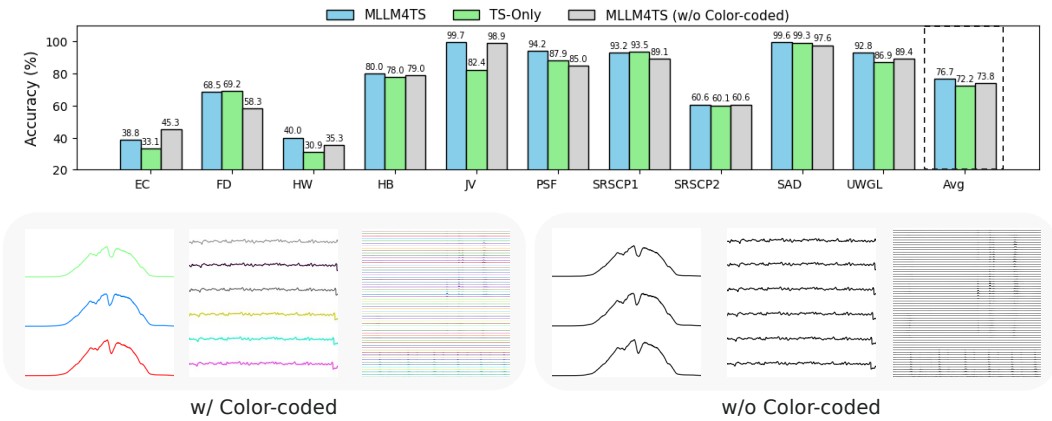

Figure 12: Impact of channel color coding on time series classification performance.

together with its robustness to hyperparameter variation (such as patch size), highlights its reliability in data-sparse environments. These characteristics make MLLM4TS a strong candidate for deployment in decision support systems across healthcare, manufacturing, finance, and climate-related services.

**Impact on Future Research.** This work is among the first to incorporate line plots, an intuitive and widely used method for visualizing time series, into the context of multi-modal time series analysis. While the proposed framework focuses on line plot representations, its architecture can be extended to incorporate aligned image and video data. Furthermore, our investigation into the roles of visual representations and language model backbones provides valuable insights for the development of explainable and agentic time series analysis, paving the way for safer and more transparent deployment in high-stakes domains.

# E  LIMITATION AND FUTURE WORK

One limitation of MLLM4TS is the additional computational overhead introduced by the vision branch, which increases runtime during the processing of visual embeddings. Future research may explore the design of more lightweight visual frontends for rendering visual representations, with the goal of improving runtime efficiency. Another promising direction is extending MLLM4TS to handle irregularly sampled time series. Given the demonstrated benefits of visual representations, converting such data into image form may offer a more natural solution, opening new avenues for multimodal time-series analysis in this context.

