# OpenReview forum: "MLLM4TS: Leveraging Vision and Multimodal Language Models for General Time-Series Analysis"
_ICLR.cc/2026/Conference — ICLR 2026 Conference Withdrawn Submission_

### Official Review · Reviewer_wPrZ · 2025-10-23

**Soundness:** 2
**Presentation:** 3
**Contribution:** 2
**Rating:** 4
**Confidence:** 3

**Summary:**

This paper proposes MLLM4TS, a multimodal framework that leverages vision and language models for time series analysis. The approach converts multivariate time series into color-coded line plots, processes them with a frozen CLIP vision encoder, and fuses visual embeddings with time series tokens through a temporal-aware alignment strategy before feeding them to a GPT-2 language model. The framework is evaluated on classification (UEA datasets), anomaly detection (TSB-AD-M), and forecasting (ETT, Weather, etc.), demonstrating improvements over time-series-only baselines and competitive performance against task-specific models.

**Strengths:**

**S1. Novel multimodal integration for general time series analysis**

MLLM4TS is among the advanced works to systematically integrate visual representations (line plots) with language models across diverse time series tasks. The temporal-aware visual patch alignment (Section 3.2) exploits the horizontal structure of time series plots to align visual patches with temporal segments, eliminating manual patch size tuning.

**S2. Comprehensive evaluation across multiple tasks and settings**

The paper provides extensive experiments spanning 30+ datasets across three major tasks. Evaluation includes full-shot, few-shot (10% data), and zero-shot settings. MLLM4TS achieves rank 1 in anomaly detection on TSB-AD-M (0.349 VUS-PR vs. 0.312 for best baseline) and competitive forecasting performance despite not being specialized for prediction.

**S3. Thorough ablation studies validating design choices**

Section 4.2 systematically ablates image layout (horizontal vs. grid), visual encoder (CLIP vs. ResNet), fusion stage (early vs. late), and patch size sensitivity. Results show horizontal layout outperforms grid (76.7% vs. 75.2% classification accuracy), CLIP exceeds ResNet (+4.1%), and early fusion is superior. The patch size analysis (Table 3) demonstrates reduced variance with visual modality (0.56 vs. 1.13 std), validating robustness claims.

**S4. Strong few-shot and zero-shot generalization**

Figure 5 and Tables 12-13 demonstrate superior generalization over time series foundation models. In zero-shot ETTh2→ETTh1 transfer, MLLM4TS (0.499 MSE) outperforms MOMENT (0.683) and LLMTime (1.961). The 10% few-shot results show maintained performance with 90% less data, highlighting data efficiency from pretrained vision-language alignment.

**Weaknesses:**

**W1. Limited novelty in core components:** The framework combines existing techniques without fundamental innovation. CLIP-ViT for visual encoding, GPT-2 for language modeling, and line plot rendering are all standard practices. The temporal-aware alignment (Section 3.2) is essentially average pooling over vertical patches followed by interpolation. The plot projection module is a simple linear layer. Compared to concurrent work like VisionTS and Time-VLM that also use vision-language models for time series, MLLM4TS lacks distinctive technical contributions beyond applying existing components to multiple tasks.

**W2. Unfair comparison setup undermines claims:** Table 2 compares MLLM4TS against specialized forecasting models but uses "One-for-One" training (separate models per horizon) while baselines may use different protocols. The paper claims MLLM4TS is "competitive" but actually underperforms PatchTST on most datasets (Weather: 0.226 vs. 0.225 MSE, ETTh1: 0.413 vs. 0.408). Comparing a general-purpose model against task-optimized ones without controlling for training compute, model size, or protocol creates misleading impressions.

**W3. Computational overhead not adequately addressed:** Figure 6 shows MLLM4TS requires 0.6s/iter inference vs. 0.1s for OFA—a 6× slowdown. Table 8 reports training time but lacks wall-clock comparisons, GPU memory usage, or throughput analysis. For a framework claiming to be "general-purpose," deployment feasibility is critical.

**W4. Inconsistent results across tasks weaken generalization claims:** While MLLM4TS excels at anomaly detection (+5.3% over OFA), improvements are marginal or negative on other tasks. In classification, the average gain is +4.5% but individual datasets show mixed results (Table 8): JV (+17.3%), but FD (-0.7%), HW (+9.1%). In forecasting (Table 2), MLLM4TS underperforms OFA on Solar (0.188 vs. 0.229 MSE) and ETTh1 (0.408 vs. 0.426 MSE). The paper provides no analysis of when/why visual modality helps. Are improvements task-dependent, dataset-dependent, or simply noise?

**W5. Vision modality contributions not clearly isolated:** The paper claims visual representations provide "global contextual information" but lacks evidence. The ablation removes the entire vision branch (Table 14-15) but does not test: (1) using only global image features (CLS token) vs. full patches, (2) comparing against simple statistical features (mean, variance, FFT) that capture global trends, (3) visualizing what visual features are learned. The color-coding ablation (Figure 12) shows only marginal differences between color-coded and grayscale plots, suggesting visual benefits may come from plot structure rather than channel encoding.

**W6. Language model necessity remains questionable:** Section 4.3 claims "language modeling capabilities are beneficial" but results are ambiguous. Table 4 shows LLM2Attn (single attention layer) matches or exceeds full GPT-2 on forecasting (0.225 vs. 0.231 MSE). The multimodal setting shows larger gaps, but this could result from CLIP's alignment with language embeddings rather than language modeling. Figure 8 demonstrates that scaling from GPT-2 (124M) to Qwen3 (1.7B) provides no gains, contradicting the "language modeling" narrative.

**W7. Missing analysis of failure cases and limitations:** The paper provides no discussion of when MLLM4TS fails. Which datasets show degraded performance with vision? Are there time series characteristics (high noise, irregular sampling, high dimensionality) where line plots hurt? Table 17 shows dimensionality reduction improves Traffic/ECL forecasting but lacks analysis of the reduction strategy (correlation-based pruning) or its impact on other tasks. How does performance vary with missing channels or incomplete plots?

**W8. Experimental protocol inconsistencies:** Different tasks use different protocols: classification uses cross-validation (Section 4.1.1), anomaly detection uses fixed train-test splits (TSB-AD-M), and forecasting uses "One-for-One" training. The few-shot experiments (Table 12) use 10% data but report results on only 2 datasets (Weather, ETTh1), while zero-shot (Table 13) tests only ETTh2→ETTh1 transfer. Why not evaluate few-shot across all 5 forecasting datasets or test multiple zero-shot transfer pairs?

**Questions:**

**Q1. What is the computational cost breakdown?**

How much overhead comes from vision encoding vs. plot rendering vs. LLM processing? Can you report GPU memory, wall-clock time, and throughput for each component?

**Q2. When does visual modality help vs. hurt?**

Provide analysis of dataset characteristics (dimensionality, length, noise level, periodicity) correlated with vision branch effectiveness. Which tasks/datasets benefit most?

**Q3. Why does scaling language models not help?**

Figure 8 shows Qwen3 performs worse than GPT-2. Is this overfitting, distribution mismatch, or fundamental limitation? Does this question the "language modeling" hypothesis?

---

### Official Review · Reviewer_LRTt · 2025-10-23

**Soundness:** 3
**Presentation:** 3
**Contribution:** 2
**Rating:** 4
**Confidence:** 5

**Summary:**

Proposes MLLM4TS, a multimodal framework integrating vision and language for general time-series analysis, addressing modality gaps between continuous time-series data and discrete language via color-coded line plots and temporal-aware visual patch alignment.

**Strengths:**

- Extends existing visual time-series encoding to a multimodal LLM-driven paradigm, unifying diverse tasks (classification, anomaly detection, forecasting) instead of task-specific designs.
- Provides comprehensive experimental details, pseudocode, and supplementary results, ensuring high reproducibility.

**Weaknesses:**

- The core idea of encoding time series as visual plots was previously proposed in [1], reducing the novelty of its visual modality design.
- Dimensionality reduction for high-channel data may discard critical cross-channel dependencies, despite claims of representative subset effectiveness.
- Baseline comparisons lack some latest multimodal time-series models.
- The claim that GPT-2 is sufficiently expressive lacks rigorous validation against diverse modern LLMs [2].

[1] Time series as images: Vision transformer for irregularly sampled time series, NeurIPS 2023
[2] Does Multimodality Lead to Better Time Series Forecasting, Arxiv 2025

**Questions:**

See the weaknesses.

---

### Official Review · Reviewer_tAm3 · 2025-10-27

**Soundness:** 3
**Presentation:** 3
**Contribution:** 2
**Rating:** 4
**Confidence:** 4

**Summary:**

The paper presents MLLM4TS, a multimodal framework for general time series analysis. It converts multivariate time series into color-coded line chart representations and leverages pre-trained vision-language models for modeling. A Temporal-Aware Visual Patch Alignment strategy is introduced to fuse temporal and spatial information, capturing both local dynamics and global dependencies. Experiments on classification, anomaly detection, and forecasting tasks show that multimodal integration substantially improves generalization and robustness.

**Strengths:**

1. The authors incorporate vision-based approaches to address the gap between continuous time series and discrete LLM tokens, showing a novel perspective.
2. The work is relatively comprehensive, as experiments are conducted across multiple downstream tasks.

**Weaknesses:**

1. The author didn't provide enough support to choose Line plots instead of other transfer methods, like the heat map.
2. The model does not seem to achieve strong performance in the domains of time series forecasting and anomaly detection.

**Questions:**

1. Using the line plot to represent the time series will cause the following concerns
   1. When the dataset contains only a few channels, the use of colored line plots results in large blank regions with limited visual information.
   2. The visibility of temporal variations in the time series is highly dependent on the selected y-axis range, which is substantially influenced by the number of channels.
   3. In the vision field, near pixels will have stronger relationships; thus, the stacking of multiple line plots requires choosing the adaptive order among them.
   4. Use color to also introduce additional information by manually setting.
2. This model includes a Large Language Model and also an additional Vision Encoder, Output layer, etc, which increases the concern about efficiency. I encourage the authors to compare their model with other light time series models like CycleNet, DLinear, SVTime, etc.

---

### Official Review · Reviewer_H2Wp · 2025-11-01

**Soundness:** 2
**Presentation:** 2
**Contribution:** 2
**Rating:** 2
**Confidence:** 4

**Summary:**

This paper proposes MLLM4TS, a multimodal framework for time series analysis based on an LLaVA-style architecture.  Additionally, the authors utilize numerical data to create line images, enhancing performance. The proposed method is comprehensively evaluated on different datasets and tasks. The proposed method is comprehensively evaluated on several datasets. However, the novelty is limited, and further discussion of related prior work is needed.

**Strengths:**

- The proposed MLLM4TS,  LLaVA-style framework,  is a good attempt at applying VLMs to time series forecasting, and serves as a meaningful extension to previous similar works.

- The authors conduct lots of experiments to evaluate the proposed method.

**Weaknesses:**

### **1. Novelty and Related Work**

The proposed framework is largely **LLaVA-style**, following the same multimodal pipeline that combines a **frozen CLIP vision encoder** with a **pretrained LLM backbone** through a projection layer to fuse multimodal features.  However, the authors claim some common strategies, such as adding a projection, combining different features, and feeding them into the LLM as their contribution. I think this strategy is widely used in VLMs and is not novel.

Additionally, the authors proposed converting time series into color-coded **line-plot images** and feeding them into a vision encoder.   It is very similar to [1], which also uses line-plot visualization, color coding, and a pretrained vision encoder to extract features from time series. The current work introduces an LLaVA-style framework, but the underlying fusion logic and modality alignment remain unchanged, similar to LLaVA and [1], which both use CLIP as a vision encoder.

Moreover, the authors should provide a detailed comparison and discussion with related works [1–5].  Currently, the claimed novelty is limited, as most components—including the CLIP encoder, which converts numerical data into an image, the multimodal fusion design- are directly adapted from existing multimodal learning frameworks.



### **2. Dataset**

The authors evaluate forecasting experiments that cover only a limited set of datasets while omitting widely used benchmarks such as ETTm1, ETTm2, and M4.  The authors should also consider the experimental setting[6].  The horizon length is 7.5 times longer than the context length, which lacks practical value.  I do not think predicting 336/720 based on 96 is a really useful setup. Humans never predict the weather based on the past 16 hours to forecast one week.


### **3. Few-shot Learning and Experimental Settings**

The few-shot forecasting evaluation is **very limited**, as which are conducted on only **two datasets**.  In addition, the choice of baselines and datasets appears inconsistent across experiments.


[1] Teaching Time Series to See and Speak: Forecasting with Aligned Visual and Textual Perspectives

[2] GEM: Empowering MLLM for Grounded ECG Understanding with Time Series and Images

[3] Time-VLM: Exploring Multimodal Vision-Language Models for Augmented Time Series Forecasting

[4] Multi-Modal View Enhanced Large Vision Models for Long-Term Time Series Forecasting

[5] Visual Instruction Tuning

[6] https://neurips.cc/virtual/2024/108471

**Questions:**

Please refer to Weakness.

---

### Note · Authors · 2025-11-14

I have read and agree with the venue's withdrawal policy on behalf of myself and my co-authors.